# The impact of fluctuations and correlations in droplet growth by collision-coalescence revisited. Part II: Observational evidence of gel formation in warm clouds.

**L. Alfonso[1], G. B. Raga[2] and Darrel Baumgardner[3]**

[1]Universidad Autónoma de la Ciudad de México, México City, 09790 México

[2]Centro de Ciencias de la Atmósfera, UNAM, México City, 04510 México

[3]Droplet Measurement Technologies, Boulder, CO, USA

Correspondence to: L. Alfonso (lesterson@yahoo.com)

**Abstract**

In recent papers (Alfonso et al., 2013; Alfonso and Raga, 2017) the sol-gel transition was proposed as a mechanism for the formation of large droplets required to trigger warm rain development in cumulus clouds. In the context of cloud physics, gelation can be interpreted as the formation of the "lucky droplet" that grows by accretion of smaller droplets at a much faster rate than the rest of the population and becomes the embryo for raindrops. However, all the results in this area have been theoretical or simulation studies. The aim of this paper is to find some observational evidence of gel formation in clouds by analyzing the distribution of the largest droplet at an early stage of cloud formation, and to show that the mass of the gel (largest drop) is a mixture of a Gaussian and a Gumbel distributions, in accordance with the pseudo-critical clustering scenario described in Gruyer et al. (2013) for nuclear multi-fragmentation.

## 1. Introduction

A fundamental, ongoing problem in cloud physics is associated with the discrepancy between the times modeled and observed for the formation of precipitation in warm clouds. Observational studies show that precipitation can develop in less than 20 minutes. For example, in Göke et al. (2007), an analysis of radar observations in the framework of the Small Cumulus Microphysics Study (SCMS),

demonstrated that maritime clouds increased their reflectivity from -5 dBZ to +7.5 dBZ in a

characteristic time of 333 s. Simulations of the collision and coalescence process, such as those

described in the review published by Beard and Ochs (1993), require longer times for precipitation

formation, unless giant nuclei (aerosols with diameters greater than 2 μm) are incorporated in the

simulation.

Numerous mechanisms have been proposed to close the gap between observations and simulations.

Some theories explain this phenomenon as an increase in collision efficiencies due to turbulence

(Wang et al., 2008; Pinsky et al., 2004, 2007, 2008), turbulence-enhanced collision rate of cloud

droplets (Falkovich and Pumir, 2007; Grabowski and Wang, 2013), and turbulent dispersion of cloud

droplets (Sidin et al., 2009).

More recent papers (Onishi and Seifert, 2016; Li et al, 2017; Li et al, 2018, and Chen et al., 2018) also

investigated the effect of turbulence in early development of precipitation.

Other research points to the supersaturation fluctuations resulting from homogeneous (Warner, 1969)

and inhomogeneous mixing (Baker et al., 1980), which allow some droplets to grow faster by

condensation in areas with larger supersaturation. Cooper (1989) found evidence of faster growth of

the larger droplets due to the variability that results from mixing and random positioning of droplets

during cloud formation. Shaw et al. (1998) explored the possibility that vortex structures in a turbulent

cloud cause variations in droplet concentration and supersaturation (at the centimeter scale), allowing

droplets in areas of higher concentration to grow more rapidly. Their calculations show an important

widening of the spectrum from this mechanism. Roach (1976) showed that the growth of larger

droplets increases from radiative cooling at the top of stratiform clouds, and by the addition of sulfate

cloud condensation nuclei (CCN) activated as droplets as a result of aqueous phase chemical reactions

(Zhang et al., 1999). In the same manner Feingold and Chuang (2000) proposed the theory that certain

organic compounds (film-forming compounds) can create a layer around droplets that inhibit their

growth, causing a fraction of droplets to grow under conditions of higher supersaturation with the

consequent widening of the spectrum. The existence of giant CCN is another of the proposed mechanisms. Even at concentrations as low as 1 per liter, they can contribute significantly to the broadening of the spectrum (Johnson 1982; Feingold et al., 1999; Yin et al., 2000; Van Den Heever and Cotton, 2007).

More recently, the sol-gel transition has been proposed as a possible mechanism for the formation of embryonic drops that trigger the formation of precipitation (Alfonso et al., 2010, 2013). Although this phenomenon is not as well known in the field of cloud physics, the sol-gel transition (also known as "gelation" in the English literature), has been widely studied in other fields of research to explain, for example, the formation of planets (Wetherill, 1990), of aerogels in aerosol physics (Lushnikov, 1978), or the emergence of giant components in percolation theory (Aldous, 1997).

In the framework of cloud physics, the sol-gel phenomenon can be interpreted as the formation of the "lucky droplet" that becomes the embryo for raindrops, and is defined by a transition from a continuous system of small droplets, to another system with a continuous spectrum plus a giant drop (runaway droplet, embryonic drop, gel) that interacts with the system increasing its mass by accretion with the smallest drops.

Telford (1955) may be the first to propose the "lucky droplet" model for collision-coalescence of cloud droplets. One of the novelties of Telford's approach was to recognize the shortcomings of the "continuous growth model", and took into account the statistical fluctuations inherent to the collision-coalescence process and its discrete nature. He performed his analysis for a cloud consisting of identical 10 µm droplets together with collector drops with twice the volume (12.6 µm radius). From this initial bimodal distribution, he found that 100 of the 12.6 µm droplets per cubic meter (a $10^{-6}$ fraction), will grow more rapidly than predicted by the continuous growth model, experiencing their first 10 coalescences after a time of approximately 5 minutes, while the time to undergo 10 collisions assuming continuous growth was about 33 minutes.

The lucky droplet model was further developed by Kostinski and Shaw (2005), who presented numerical evidence that their model can lead to a rapid development of precipitation. Their analysis was based on the derivation of the distribution of times for N collisions (which resulted to be the Erlang distribution). They concluded that the $10^{-6}$ lucky droplets are expected to reach the 50 µm 10 times faster than the average droplet. More recently, Wilkinson (2016) advanced further the model by using large deviation theory (Touchette, 2009). He derived the probability for the time T to undergo N collisions being a very small fraction of its mean value, and showed that the time scale for the initiation of precipitation is smaller than the mean time for a single collision.

The results obtained by Kostinski and Shaw (2005) were tested by Dziekan and Pawlowska (2017) by calculating the "luck factor", ie, how much faster the luckiest droplets grow to r=40 µm compared to the average droplets. They estimated that the luckiest $10^{-3}$ fraction will cross the size gap around 5 times faster, and the luckiest $10^{-5}$ fraction around 11 times faster, in good agreement with the results obtained by Kostinksi and Shaw (2005) (about 6 and 9 times faster respectively).

However, previous efforts on this direction were mainly focused on finding the distribution of times for *N* collisions (Telford, 1955; Kostinski and Shaw, 2005; Wilkinson; 2016), while we were concentrated on studying the "lucky droplet" size distribution to determine whether or not the runaway growth process due to collision-coalescence has started.

 Recent studies that address the sol-gel transition interpretation in cloud physics (Alfonso et al., 2013; Alfonso and Raga, 2017) analyze the problem from the theoretical and simulation point of view. The aim of the present work here is to find observational evidence of gel formation, taking as a reference recent studies in percolation theory (Botet and Płoszajczak, 2005) and nuclear physics (Botet et al., 2001; Gruyer et al., 2013), which can shed some light on the gel (largest droplet) size distribution during the initial stage of precipitation formation.

The paper is organized as follow: Section 2 presents an overview of previous results for both infinite and finite systems.  An analysis of the largest droplet distribution from synthetic data obtained from

Monte Carlo simulations (for the product and hydrodynamic kernels, respectively) is presented in section 3, section 4 is devoted to the analysis of experimental data, and finally, in section 5 we discuss our results accompanied by the relevant conclusions.

### 2. **An overview of previous theoretical and experimental results**

*2.1 Results for infinite systems in coagulation and percolation theory*

The most commonly accepted approach to model the collision coalescence process in cloud models with detailed microphysics relies upon the Smoluchowski kinetic equation or kinetic collection equation (KCE), governing the time evolution of the average number of particles. The discrete form of this equation can be written as (Pruppacher and Klett, 1997):

$$\frac{\partial N(i,t)}{\partial t} = \frac{1}{2}\sum_{j=1}^{i-1} K(i-j,j)N(i-j)N(j) - N(i)\sum_{j=1}^{\infty} K(i,j)N(j) \tag{1}$$

where $N(i,t)$ is the average concentration of droplets with mass $x_i$ at time t, and $K(i,j)$ is the coagulation kernel related to the probability of coalescence of two drops of masses $x_i$ and $x_j$. In Eq. 1, the first term in the r.h.s. describes the average rate of production of droplets of mass $x_i$ due to coalescence between pairs of drops whose masses add up to mass $x_i$, and the second term describes the average rate of depletion of droplets with mass $x_i$ due to their collision and coalescence with other droplets.

However, the KCE may have a serious limitation in some cases (Lushnikov, 2004) and, hence, cannot accurately describe the coagulation process. The limitation lies essentially in the fact that the coagulation equation inevitably creates particles with infinite mass. For example, for a multiplicative coagulation kernel ($K(i,j)=Cx_ix_j$), an attempt to calculate the second moment of the droplet mass spectrum:

$$M_2(t) = \sum_{i=1}^{N_d} x_i^2 N(i,t) \tag{2}$$

leads to the result:

$$M_2(t) = \frac{M_2(t_0)}{1 - CM_2(t_0)t} \tag{3}$$

$$T_{gel} = [CM_2(t_0)]^{-1} \tag{4}$$

Then, after $t=T_{gel}$ the second moment may become undefined, and the total mass of the system starts to decrease (See Appendix A for more details). This result applies to infinite (with negligible fluctuations and correlations) coagulating systems in the thermodynamic limit, which is the limit for a large number $K$ of particles where the volume $V$ is taken to grow in proportion with the number of particles. Then $K, V \rightarrow \infty, K/V \rightarrow N < \infty$. The infinite system interpretation of the sol-gel transition assumes the presence of a gel phase (which is not predicted by the KCE equation), and introduces an additional assumption as to whether or not the gel interacts with the infinite size clusters that are not described by the KCE equation.

The other scenario considers that coagulation takes place in a system with a finite number of monomers in a finite volume. This approach is based on the scheme developed by Markus (1968) and Bayewitz et al. (1974), and was studied by Lushnikov (1978, 2004), Tanaka and Nakazawa (1993, 1994) and Matsoukas (2015) by using analytical tools, and more recently by Alfonso (2015) and Alfonso and Raga (2017) numerically. Within this approach there is no mass loss, and the phase transition is manifested in the emergence of a giant particle that contains a finite fraction of the total mass of the system. Solutions in the post-gel regime were obtained analytically by Lushnikov (2004 and Matsoukas (2015), and numerically by Alfonso and Raga (2017).

The sol-gel transition has been observed experimentally, for example: aerogels in aerosol physics (Lushnikov et al., 1990), and in other theoretical models such as that of percolation (Botet and Płoszajczak, 2005; Kolb and Axelos, M. A., 1990) where there is a close analogy between percolation and droplet coagulation. In bond percolation, each lattice corresponds to a monomer, and a proportion $p$ of active bonds is set randomly between sites. Then clusters of size $s$ are defined as an ensemble of

$s$ occupied sites connected by active bonds. For a definite value of $p=p_c$, a macroscopic cluster appears, corresponding to the sol-gel transition.

Recent results in percolation theory show that the largest cluster follows the Gumbel distribution for subcritical percolation (Bazant, 2000) and, at the critical point, follows the Kolmogorov-Smirnov (K-S) distribution (Botet and Płoszajczak, 2005). The K-S distribution is the distribution of the maximum

value of the deviation between the experimental realization of a random process and its theoretical cumulative distribution and it has the cumulative distribution:

$$K_1(z) = \sum_{k=-\infty}^{\infty} \left(-1\right)^k e^{-k^2 \pi^2 z/6} \tag{5a}$$

or the equivalent expression:

$$K_1(z) = \sqrt{\frac{6}{\pi z}} \sum_{k=-\infty}^{\infty} e^{-3(2k+1)^2/(2z)} \tag{5b}$$

Botet and Płoszajczak (2005) also found evidence (from numerical solutions of the KCE equation) that, for multiplicative coalescence (with a collection kernel proportional to the product of the masses), the largest cluster follows the distribution in Eq. 5 at the time of the phase transition. At this point, a hypothesis is formed in which the results obtained in percolation are extrapolated in order to find the probability distribution of the largest (runaway) droplet at $t=T_{gel}$.

*2.2 Some theoretical and experimental results for finite systems in coagulation theory and nuclear physics.*

We will now consider some results obtained for finite systems in coagulation theory (Botet, 2011) and in nuclear physics (Gruyer et al., 2013). Unlike those in infinite systems, fluctuations and correlations in a finite system are not negligible. $N \rightarrow \infty$

We must emphasize that phase transitions cannot take place in a finite system. This is due to the fact that a phase transition is defined as a singularity in the free energy or any thermodynamic property of a system; and for finite-sized systems, the free energy is proportional to the logarithm of a finite number of exponentials, which are always positive. Consequently, those singularities are only possible

within infinite systems by taking the thermodynamic limit. Then, for finite systems, the notion of

pseudo-critical region is introduced (which is the finite system equivalent of a sol-gel transition time). Some interesting simulation and experimental results were obtained for these systems in Botet (2011) for the Smoluchowski model (1) and in Gruyer et al., (2013) for nuclear multi-fragmentation. Botet et al. (2001) found, from stochastic simulations of coagulation process with the product kernel (for a system of $N$=512 monomers), that the distribution of the largest cluster in the pseudo-critical region

can be described as a mixture of the well-known Gaussian and Gumbel distributions:

$$f(x,\theta,\mu_1,\beta,\mu_2,\sigma) = \theta Gumbel(x,\mu_1,\beta) + (1-\theta)Gauss(x,\mu_2,\sigma) \tag{6}$$

In Eq. 6, the coefficients $\theta$ and (1- $\theta$) are the mixture weights (probabilities associated with each component). The individual distributions $Gumbel(x,\mu_1,\beta)$ and $Gauss(x,\mu_2,\sigma)$ are the mixture components.

The Gumbel distribution is one of the asymptotic distributions from Extreme Value Theory (EVT) and has the form:

$$Gumbel(x,\mu,\beta) = e^{-e^{-(x-\mu)/\beta}} \tag{7}$$

where $\mu$ is the position parameter and $\beta$ the scale parameter. The distribution in Eq. 6 has its origin in the fact that, for finite systems, in the pseudo-critical zone, the system experiences large fluctuations

and the gel distribution is a combination of both distributions, a Gumbel and a Gaussian (Gruyer et al, 2013). A similar result was obtained by Botet (2011) using synthetic data from stochastic simulations, for collision probabilities proportional to the product of the masses.

The fundamental hypothesis of our work is that the gel mass (largest drop) in the initial phase of precipitation formation, is distributed as a mixture of two asymptotic distributions: one Gumbel and

one Gaussian, following the pseudo-critical clustering scenario described in Gruyer et al. (2013).

### 3. Analysis of the largest droplet distribution obtained from synthetic data.

*3.1 Results for the product kernel (K(i,j)=Cxᵢxⱼ)*

For synthetic data analysis, the empirical distributions of the largest droplet mass ($M_{max}$) were obtained from Monte Carlo simulations, following Botet (2011). The species accounting formulation (Laurenzi et al., 2002) of the stochastic simulation algorithm (SSA) of Gillespie (1975) that rigorously accounts for fluctuations and correlations in a coalescing system was used for the stochastic simulation in this work (See Appendix B).

The main difference between the Gillespie's SSA and other Monte Carlo methods based on the simulation particles (SIP) approach (like the Super Droplet method developed by Shima et al., (2009)), is that the Gillespie's SSA involved the collision of only two physical particles (droplets in our case) per MC cycle, while in the approach based on SIP in each MC cycle collide SIP (super-droplets, for example) that represents a multiple number of droplets with the same attributes (radius $r$ or mass in

the simplest case) and position. However, Gillespie's SSA works perfectly for our purposes, because, due to the finiteness of our systems, our simulations are performed for small volumes with a small number of droplets (in the range 50-300 cm$^{-3}$).

Our methodology of synthetic data analysis consists in generating $N$-realizations (at each time step) using the algorithm of Gillespie. For each realization, there is a certain distribution of droplets. The

largest droplet mass obtained from each distribution at each realization (for a fixed time step) would be the distribution to be fitted to the distribution in Eq. 6. Then, the sample size would be equal to the number of realizations of the Monte Carlo algorithm.

Simulations were performed for the product kernel ($K(i,j)=Cx_ix_j$), with an initial mono-disperse distribution of 100 droplets of 14 µm in radius (droplet mass $1.15\times10^{-8}$g) in a cloud volume of 1 cm$^3$,

with C= $5.49x10^{10}$ cm$^3$ s$^{-1}$.

The product kernel is proportional to the product of the masses of the colliding droplets. It is widely used because analytical solutions of the KCE or Smoluchowski equation (Eq. 1) have been obtained tor this kernel Golovin (1963), Scott (1968), Drake (1972) and Drake and Wright (1972). The value of

the constant C (C=5.49×10$^{10}$) in the product kernel is the result of the polynomial approximation

$K(x, y) = A + B(x + y) + Cxy$ (Long, 1974) of the hydrodynamic collection kernel (Eq. 11).

The empirical distribution of the maxima was obtained for 1000 realizations of the stochastic algorithm. 


Figures 1(a)-1(d) present the largest droplet mass empirical distributions obtained at four different times. Note that Eq. 6 provides a good fit for the distribution of the mass of the largest droplet ($M_{max}$) both around and far from the sol-gel transition time ($T_{gel}$), which was calculated from Eq. 4 and found

equal to 1378 s.

Figure 2 presents the time evolution of the coefficient $\theta$, which represents the mixing fraction in Eq. 6, for the time interval [500s, 2000s]. Despite the noisy behavior of the coefficient $\theta$ (due to the finiteness of the system), there is a decreasing trend with time, showing larger values of $\theta$ (~0.65) for times close to 500s and values down to 0.2 at the end of the time interval. This figure indicates that,

although the largest droplet distribution is adequately described by a mixture of Gaussian and Gumbel distributions, it has a larger Gumbel component (see Eq.6) during the early stages of the coagulation process. As time progresses, the Gaussian contribution becomes more important (smaller values of $\theta$) in providing a better fit to the largest droplet mass distribution.

These findings are in accordance with Gruyer et al. (2013) and Botet (2011): at an early stage of

coagulation development, correlations are negligible, and consequently, the largest fragments can be considered independent random variables. Therefore, the probability distribution of the largest fragment is given by the Limit Theorem for Extremal Variables, which states that the maximum of a

sample independent and identically distributed random variables can only converge in distribution to one of three possible distributions: Gumbel, Fréchet or Weibull.

As the coagulation process continues, fluctuations and correlations between droplets increase and the system reaches a critical point (Alfonso and Raga, 2017), where the largest droplets are no longer independent random variables, the Limit Theorem for Extremal Variables no longer applies, and the largest droplet distribution is no longer described by a Gumbel distribution. At later times, away from the pseudo-critical region, the Gaussian contribution is the most important part of the largest droplet

mass distribution. This can be explained by the additive nature of the process at this stage (Botet, 2011; Gruyer et al. 2013; Clusel and Bertoin, 2008), and the central Limit Theorem applies.

In the intermediate zone (which can be defined as the pseudo-critical zone), the distribution is well described by a mixture of Gumbel and Gaussian distributions and the weights associated with each distribution are comparable. It would be expected to observe $\theta = 0.5$ at the infinite system critical

point, $T_{gel}$ , found to be 1378s from Eq. (4). However, due to the finiteness of the system, the critical point corresponds approximately to a value $\theta = 0.35$ (see Fig. 2).

We can find whether or not a system is in the pseudo-critical region by defining the following ratio (Botet, 2011; Gruyer et al., 2013):

$$\eta = \frac{w_{Gaussian} - w_{Gumbel}}{w_{Gaussian} + w_{Gumbel}}$$ (8)

where $w_{Gumbel} = \theta$ and $w_{Gaussian} = 1 - \theta$ are the relative weights of the Gumbel and Gaussian distributions, respectively (see Eq. 6). By definition $\eta = +1, -1$ corresponds to pure Gaussian and Gumbel distributions. If $-1 < \eta < 1$ the system is in the pseudo-critical domain.

Alternatively, Botet (2011) estimates the limits of the pseudo-critical region as the times where the largest droplet mass standard deviation $\sigma(M_{max})$ calculated from Eq. 9 is small.

$$\sigma(M_{max}) = \sqrt{\frac{1}{N_r} \sum_{i=1}^{N_r} (M_{max}^i - \langle M_{max} \rangle)^2}$$ (9)

In Eq. (9), $N_r$ is the number of iterations of the stochastic simulation algorithm of Gillespie (1975), $M_{max}$ the mass of the largest particle and $\langle M_{max} \rangle$ its ensemble mean over all the realizations.

Even though the second moment of the distribution $M_2(\tau)$ diverges (see Eq. 3) for the infinite system, there is no divergence of the second moment for a finite system (with no critical behavior). For that case, the standard deviation for the largest particle mass ($\sigma(M_{max})$) is expected to reach a maximum in the vicinity of $T_{gel} = [CM_2(t_0)]^{-1}$. Moreover, computing the time evolution of the normalized standard deviation $\sigma(M_{max})/\langle M_{max} \rangle$ instead of $\sigma(M_{max})$ yielded better results in estimating $T_{gel}$ in Inaba (1999), Alfonso et al. (2008, 2010, and 2013) and Alfonso and Raga (2017).

Figure 3a shows the time evolution of $\sigma(M_{max})/\langle M_{max} \rangle$ as an example, for the system defined at the beginning of this section. Note that the maximum occurs at $T$=1315 s, close to $T_{gel}$=1378 s calculated from Eq. (4), and the time when the maximum of $\sigma(M_{max})/\langle M_{max} \rangle$ occurs is a reliable estimate of the sol-gel transition time for the corresponding infinite system.

Botet (2011) defines $\sigma = 0.1\sigma_{max}$ as the limits of the pseudo-critical interval, which corresponds to $t_{inf} = 0.37 T_{gel}$ and $t_{sup} = 1.5 T_{gel}$ (see Figure 3b). While Eq. (8) could be used to determine if a sample collected inside a cloud is in the pseudo-critical region, Eq. (9) implies that the time evolution of $\sigma(M_{max})$ is needed and therefore, a practical application is only viable in the case of synthetic data obtained from stochastic simulations, or cloud droplet data collected dynamically at different times or cloud levels.

*3.2. Numerical results for turbulent conditions*

In our simulations, turbulent effects were considered by implementing the turbulence induced collision enhancement factor $P_{Turb}(x_i, x_j)$ that is calculated in Pinsky et al. (2008) for a cumulonimbus with dissipation rate $\varepsilon$=0.1 m$^2$s$^{-3}$and Reynolds number Re$_\lambda$=2×10$^4$, and for cloud droplets with radii $\leq$21µm. The turbulent collection kernel has the form:

$$K_{Turb}(x_i, x_j) = P_{Turb}(x_i, x_j)K_g(x_i, x_j) \tag{10}$$

where $K_g(x_i, x_j)$ is hydrodynamic kernel, which considers collisions between droplets under pure gravity conditions and has the form:

$$K_g(x_i, x_j) = \pi(r_i + r_j)^2 |V(x_i) - V(x_j)| E(r_i, r_j) \tag{11}$$

The hydrodynamic kernel takes into account the fact that droplets with different masses ($x_i$ and $x_j$ and corresponding radii, $r_i$ and $r_j$) have different terminal velocities $V(x_i)$, which are functions of their masses. In Eq. 10, $E(r_i, r_j)$ are the collection efficiencies calculated according to Hall (1980).

Monte Carlo simulations were performed with an initial bi-modal distribution (200 droplets of 10 μm in radius, and 50 droplets of 12.6 μm) for a cloud volume of 1 cm$^3$.

As we want to perform simulations for small systems (with a small number of particles) for which fluctuations and correlations are relevant, the number of droplets per cm$^3$ use in the simulations are small. They are of the same order of the droplet concentrations for each block obtained from observations, which fluctuate between 0 and 392 cm$^{-3}$, with an average of 146 cm$^{-3}$(see Fig.6).

The empirical distribution for the largest droplet mass was generated by extracting the maximum from the droplet distribution at each realization, for a fixed time step. Additionally, the ratio $\sigma(M_{max})/\langle M_{max}\rangle$ is evaluated from 1000 realizations of the Monte Carlo algorithm (see Fig. 4), that reaches its maximum at around 1815 s, and serves as an estimate for the sol-gel transition time for the infinite system. Four empirical probability distributions were fitted to the combined distribution (Eq. 6) for times in the vicinity of $T_{gel}$. The results are displayed in Figures 5(a)-5(d). Note that also for this case, the combined distribution (Eq. 6) provides a good fit for the largest droplet mass. Moreover, the coefficient $\theta$ decreases in time (check Fig. 5), in concordance with section 3.1.

4. **Analysis of the largest droplet (gel) radius distribution from observations.**

In this section, the methodology of analysis described before is applied to a dataset of cloud droplet size distribution (2-50 µm) collected with a Droplet Measurement Technologies fog monitor (FM-120)
installed on a hilltop in Are, Sweden. The FM-120 is a single particle optical spectrometer (Spiegel et al., 2012) that derives size from light scattered from individual droplets that pass through a focused laser beam. The equivalent optical size ranges from 2-50 µm. The fog monitor sample volume has a cross sectional area of 0.25 mm$^2$ and a flow speed of 14 m/s. The raw data consists of each droplet's radius and inter-arrival time (elapsed time since previous particle). More than seven million droplets
were processed over a period of 4 hours.

The block maxima (BM) approach in extreme value theory (EVT) was applied, which requires dividing the observation period into non-overlapping periods of equal size and restricts attention to the maximum observation in each period [see Gumbel (1958)].

Following the BM approach, considering the sectional area and flow speed, the time series was divided
into consecutive unit blocks of 1cm$^3$ in volume, corresponding to a cloud length of approximately 400 cm (~0.3 s interval in the time series). The droplet distributions in each unit block, are equivalent to the distributions obtained for each realization (for a fixed time) of the Monte Carlo algorithm described in the previous section, and each block can be interpreted as an independent realization of a stochastic process.

The maximum (radius of the largest droplet) is recorded from each consecutive unit block in order to generate the distribution for comparison with the theoretical combined distribution described in Eq 6. The sample size corresponds to the number of consecutive blocks in which the time series was divided, which in this case is 49647 blocks which is equivalent to about 4 hours of data. Figure 6 displays the number of droplets in each block, which fluctuate between 0 and 392, with an average of 146. Since
each block is considered as a realization of a random process, the largest droplet radius series must be fitted to the combined distribution in Eq. 6 for samples with certain conditions of homogeneity.

The average sample size (number of unit blocks) for which the largest droplet maxima can be fitted to the combined distribution in Eq. 6 is then estimated. This expected value can be calculated from the following procedure:

The conditional probability $P(Admixture|x)$, where $x$ is the sample size, is calculated using Monte Carlo simulations. This calculation uses a given number of consecutive blocks with a mixture of distributions. The simulations are carried out by randomly choosing $N_{total}$ samples from the measurements (that consist of consecutive blocks) of size $x$, fitting the data to the distribution in Eq. 6, and determining if they do or do not follow that distribution. The decision is based on application

of the Kolmogorov-Smirnov (K-S) goodness of fit test for a confidence level $\alpha = 0.05$. The experimental statistics for the K-S test can be obtained by arranging the data in ascending order ($x_1$, $x_2, ..., x_n$), and deriving the maximum difference between the rank statistics (i-1)/n and the theoretically calculated cumulative density function $F(x_i)$:

$$D_n = \max_{1 \leq i \leq n} \left( \max \left| F(x_i) - \frac{i-1}{n} \right|, \max \left| \frac{i}{n} - F(x_i) \right| \right) \tag{12}$$

If this value of $D_n$ is smaller than a certain threshold value $D_n^\alpha$, we accept that the data obeys the probability distribution under consideration and the null hypothesis $H_0$ cannot be rejected at a significance level α. The significance level α refers to the probability of the assumed distribution pattern being rejected. The limiting values of $D_n^\alpha$ can be calculated from the K-S cumulative distribution (See Eqs. 5a and 5b). Tables with limiting values can be found, for example in Gnedenko

360 (2017).

However, given that the parameters of the distribution $F(x)$ were estimated from the observed data, theoretical limiting values provided by the K-S cannot be used. In this case, the limiting values $D_n^\alpha$ are smaller than the case with known parameters and must be obtained via Monte Carlo simulations (See Appendix C for more details). Then, the conditional probability can be calculated as:

$$P(Admixture|x) = N_0/N_{total} \qquad (13)$$

where $N_0$ is the number of cases for which the null hypothesis $H_0$) at $\alpha$=0.05 cannot be rejected. However, what is really needed is the conditional probability $P(x|Admixture)$, that is the probability that a sample has size $x$, given that the data (viewed as a time series of maxima for each block) in that sample follow a mixture of distributions. This probability can be calculated using Bayes' theorem from the expression:

$$P(x|Admixture) \propto P(Admixture|x)\pi(x) \qquad (14)$$

By writing this theorem in the form (13), we are assuming that the marginal likelihood is considered as a normalization factor. Therefore, $P(x|Admixture)$ can be computed using expression (14) and then normalized under the requirement that it is a probability mass function (pmf). In (14), the prior probability $\pi(x)$ is assumed to have a uniform distribution. Then, the expected value $\langle x \rangle$ can be calculated from the expression:

$$\langle x \rangle = \sum P(x|Admixture)x \qquad (15)$$

Turning to a concrete example, $N_{total}$=100 samples with sizes $x$=100, 200,…, 1000 were randomly selected from the data; and the probability $P(Admixture|x)$ calculated following (13). The probability mass function $P(x|Admixture)$ (pmf) was obtained by applying the procedure previously described and the expected value was found to be $\langle x \rangle = 544$ (about 163 s).

A thorough statistical analysis was conducted by fitting $M_{max}$ to the combined distribution in Eq. 6 for 100 samples with sizes at and below the average (100, 200, 300, .., 500) that were randomly selected from the entire dataset (49647 blocks). For each random sample three null ($H_0$) hypotheses were verified: i) the sample comes from a mixture of distributions (6): ii) the sample comes from a Gumbel distribution; iii) the sample comes from a Gaussian distribution. The three hypotheses were examined by the K-S method with limiting values calculated from Monte Carlo simulations (see Table C1).






The results for sample sizes 100, 200, 300, 400 and 500 are shown in Table 1. As an example, for case 1 (sample size 100) the null hypothesis $H_0$ at α=0.05 was rejected for 13, 35 and 92 samples for the

mixture, Gaussian and Gumbel distributions, respectively. For case 2 (sample size 200), the null hypothesis was rejected for 27, 58 and 96 samples. For $n$=500 for the mixture of distributions (6), the null hypothesis $H_0$ was rejected for 50 samples. For the Gumbel distribution, the null hypothesis was rejected for all the samples (100) and the null hypothesis for the Gaussian distributions was rejected for 83 samples.

The results shown in Table 1, confirm that for all sample sizes, the mixture of distributions provides a better fit than the Gumbel and Gaussian distributions, confirming the correctness of the choice of the mixture of distributions (Eq. 6) for modelling the largest droplet radius. As an example, Figs. 7a-d present, for a sample size of $n$=500, the largest droplet mass empirical distributions obtained for four different samples that are distributed following the mixture, and the corresponding fit of Eq. 6.


## 5. Discussion and conclusions

An infinite system has two possible evolutionary phases: the ordered phase and the disordered or statistical phase. In the disordered phase there is a continuous droplet distribution and a near-equality of the largest and second largest mass. After the sol-gel transition, there is an ordered phase

characterized by the existence of a giant macroscopic droplet (gel) coexisting with an ensemble of microscopic particles.

A finite system can be in the ordered, disordered and pseudo-critical phases, according to the scenario described in Botet (2011) and Gruyer et al. (2013). The ratio $\eta$, defined in Eq. 8, takes values between $\eta = +1, -1$, which correspond to pure Gaussian and Gumbel distributions, and when $-1 < \eta < 1$ the

system is in the pseudo-critical domain. In the disordered phase, fluctuations and correlations are negligible, there are only a few collision events, and $M_{max}$ is the largest part of randomly distributed droplets. In that case, the distribution of the mass of the largest droplets follow a Gumbel distribution.

At later times in the evolution of the finite system, there are many collision events and $M_{max}$ is the result of the coalescence of other droplets. There is an additive process, the central limit theorem applies and the mass (or radius) of the largest droplets follows a Gaussian distribution.

In the pseudo-critical phase, the fluctuations and correlations are no longer negligible and the distribution is of neither one nor the other asymptotic forms (Gumbel or Gaussian). In this case, the fit of the largest droplet mass (gel), is a mixture of a Gumbel (disordered state) and Gaussian (ordered state) distributions. As was demonstrated in the preceding section, this combined distribution (Eq. 6) is a good approximation to the largest droplet distribution (gel) in the pseudo-critical region. The fact that the mixture of distributions provides a better fit than the Gumbel and Gaussian distributions shows that the samples selected in our study are mainly in the pseudo-critical phase. To confirm this fact, the ratio $\eta$ was calculated for 1000 samples of size $n=500$ selected randomly from the data. Figure 8 shows that for 90% of the samples the ratio $\eta$ lies in the interval [-0.9, 0.9], clearly indicating that samples are in the pseudo-critical region.

We could show that the gel radius (largest droplet) is well described as a mixture of the two asymptotic distributions, because the effect of the collision-coalescence process was in some way isolated for the orographic cloud data analyzed in this report. A similar analysis could be performed for the early stage of a convective cloud formation, before some other processes, e.g. entrainment, mixing, turbulence or ice formation, could obscure the finite system pseudo-critical scenario, and the gel formation that is basically a consequence of the collision-coalescence process could no longer be observed.

In this work, the early stage of formation of a warm cloud is viewed in the context of critical phenomena theory, and can be thought of as being in ordered, disordered or pseudo critical phases. The disordered phase corresponds to a cloud with a droplet spectrum formed mainly by the cloud condensation nuclei activation process, with an almost random distribution of particles, and the distribution of the mass of the largest droplets is Gumbel. In the pseudo-critical phase a giant droplet (gel) locally coexists with a continuous ensemble of small droplets. As the system considered is finite,

there is no sudden change from disordered to ordered phase (sol-gel transition), but rather there is a pseudo-critical phase in which fluctuations are important and the gel distributes according to Eq. 6.

The analysis presented here of the largest droplet distribution provides useful insight into the early stages of cloud development in warm clouds. In follow up studies, the analysis of cloud data at different time or distance from cloud base would be helpful in identifying the pseudo-critical phase and tracking the transition from the disordered to the ordered phase dynamically.

*Acknowledgements:* This study was funded by a grant from the Consejo Nacional de Ciencia y

Tecnología de Mexico (SEP-Conacyt) CB-284482.

### 6. Appendix A.

The sol-gel transition time $T_{gel}$ is defined as the time when the second moment $M_2(t) = \dfrac{M_2(t_0)}{1 - CM_2(t_0)t}$

becomes infinite, then $1 - CM_2(t_0)t = 0$, and $T_{gel} = [CM_2(t_0)]^{-1}$. The equation for $M_2(t)$ (moment of

order 2 with respect to mass) can be found from the general equation for moment evolution that was obtained by Drake (1972) from the continuous form of the kinetic collection equation (1). It has the form:

$$\frac{dM_n(t)}{dt} = \frac{1}{2}\int_0^\infty\int_0^\infty \left[(x+y)^n - x^n - y^n\right]K(x,y)N(x,t)N(y,t)dxdy \qquad (A1)$$

In (7), *K(x,y)* is the collection kernel, *N(x,t)* is the average droplet concentration and *x* is the droplet

mass. If we consider the product kernel $K(x,y) = C(xy)$ in (A1), then, the equation for the second moment is:

$$\frac{dM_2(t)}{dt} = C[M_2(t)]^2 \qquad (A2)$$

With solution $M_2(t) = \dfrac{M_2(t_0)}{1 - CM_2(t_0)t}$

After $T_{gel}$, a runaway droplet forms, and the kinetic collection equation is no longer valid, since the assumption of a continuous distribution breaks down. There is in essence a phase transition in the system from a continuous distribution to a continuous distribution *plus* a runaway droplet.

### 7. Appendix B: The Monte Carlo algorithm.

In this study, the species accounting formulation (Laurenzi et al., 2002) of the stochastic simulation algorithm (SSA) of Gillespie (1975) was used for the stochastic simulation. The steps below summarize the algorithm:

1) **Initialization (set initial numbers of species, set $t$=0, set stopping criteria):** Initialize the number of droplets in each species (the species are defined as droplets of different sizes). There is a unique index $\mu$ for each pair of droplets $i, j$ that may collide. For a system with $N$ species

$$\left( n_1, \quad n_2, \quad ... \quad , n_N \right) \quad \mu \in \frac{N(N+1)}{2}.$$ The set $\{\mu\}$ defines the total collision space, and is equal to the total number of possible interactions.

2) **Monte Carlo step: D**etermine the next collision to occur and the time to the next collision.

The next collision $\mu$ is calculated according to the distribution $P(\mu) = \dfrac{a_\mu}{\alpha}$, from the inequality:

$$\sum_{v=1}^{\mu-1} a_v < r_2 \alpha \leq \sum_{v=1}^{\mu} a_v \tag{B1}$$

Where $r_2$ is a uniformly distributed random number in the interval (0,1), $a_\mu$ are calculated from the probabilities:

$a(i, j) = V^{-1} K(i, j) n_i n_j dt \equiv \mathrm{Pr}\{$ Probability that two unlike particles $i$ and $j$ with populations (number of particles) $n_i$ and $n_j$ will collide within the imminent time interval$\}$

$$a(i,i) = V^{-1}K(i,i)\frac{n_i(n_i-1)}{2}dt \equiv \Pr\{ \text{ Probability that two particles of the same species } i$$

with population (number of particles) $n_i$ collide within the imminent time interval}

and $\alpha = \sum_{v=1}^{\frac{N(N+1)}{2}} a_v$ . As the time to the next collision is exponentially distributed with mean $1/\alpha$

(Gillespie, 1975), and that $1\text{-}r_1=r^*_1$ is a uniformly distributed random number in the interval [0,

1], then the time $\tau$ to the next collision can be calculated from the expression:

$$\tau = \frac{1}{\alpha}\ln\left(\frac{1}{r_1^*}\right) \tag{B2}$$

**3)**   Increase the time by the randomly generated time in Step 2. Change the numbers of species

to reflect the execution of a collision.

**4)**  If stopping criteria are not met, go to step 2.

## 8. Appendix C: Procedure for estimating the Kolmogorov-Smirnov goodness of fit test limiting values for distributions with unknown parameters.

When parameters of a distribution are estimated from the data, the limiting values provided for the

Kolmogorov-Smirnov criterion cannot be used. In this case, approximate limiting values and $p$-

values can be obtained via Monte Carlo simulations. First, the parameter vector

$\hat{\phi} = \left(\hat{\theta}, \ \hat{\mu}_1, \ \hat{\mu}_2, \ \hat{\beta}, \ \hat{\sigma}\right)$ is estimated for a given sample of size $n$, and the test statistics (Eq.

12) are calculated assuming that the sample is distributed according to $F\left(x;\hat{\phi}\right)$, returning a value

of $D_n$. Next, a sample of size $n$ $F\left(x;\hat{\phi}\right)$ variates is generated and the parameter vector $\hat{\phi}_1$ is

estimated. The test statistics is again calculated assuming that the sample is distributed according

to $F\left(x;\hat{\phi}_1\right)$. Such a calculation was made for different sample sizes ($n$=100, 200, …, 500) 1000

times, and the distribution pattern of $D_n$ was derived (See Table A1). Then, 5% percent point (for α=0.05) from the greater side was taken as the estimated $D_n^{\alpha=0.05}$ limiting values. The estimate of *p*-value is calculated as the relative number of occasions is which the test statistics is at least as large as $D_n$. The numerically calculated K-S limiting values for the three distributions under analysis (mixture, Gumbel and Gaussian) for α=0.05 are shown in Table 3. As can be checked in Table C1, the values are smaller than the case with known parameters, that can be estimated (for α=0.05) as $1.36/\sqrt{n}$.

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

Table 1. For each sample size, number of samples with the null hypothesis $H_0$ rejected at α=0.05 for all the distributions.

| Case | Total number of random samples | Sample size | Fitted Distributions | At α=0.05 Reject $H_0$ (Number of Samples) |
|------|-------------------------------|-------------|---------------------|-------------------------------------------|
| 1 | 100 | 100 | Mixture | 13 |
|   |     |     | Gumbel | 92 |
|   |     |     | Gaussian | 35 |
| 2 | 100 | 200 | Mixture | 27 |
|   |     |     | Gumbel | 96 |
|   |     |     | Gaussian | 58 |
| 3 | 100 | 300 | Mixture | 35 |
|   |     |     | Gumbel | 98 |
|   |     |     | Gaussian | 70 |
| 4 | 100 | 400 | Mixture | 40 |
|   |     |     | Gumbel | 100 |
|   |     |     | Gaussian | 77 |
| 5 | 100 | 500 | Mixture | 50 |
|   |     |     | Gumbel | 100 |
|   |     |     | Gaussian | 83 |




Table C1. Estimated limiting values (for α=0.05) for the Kolmogorov-Smirnov goodness of fit test for the three distributions.

| Sample size | K-S (estimated) limiting values ($D_n$) for α=0.05 | | |
|---|---|---|---|
| | Mixture | Gaussian | Gumbel |
| 100 | 0.0725 | 0.0873 | 0.0853 |
| 200 | 0.0494 | 0.0624 | 0.0630 |
| 300 | 0.0432 | 0.0517 | 0.0487 |
| 400 | 0.0369 | 0.0461 | 0.0419 |
| 500 | 0.0324 | 0.0414 | 0.0396 |



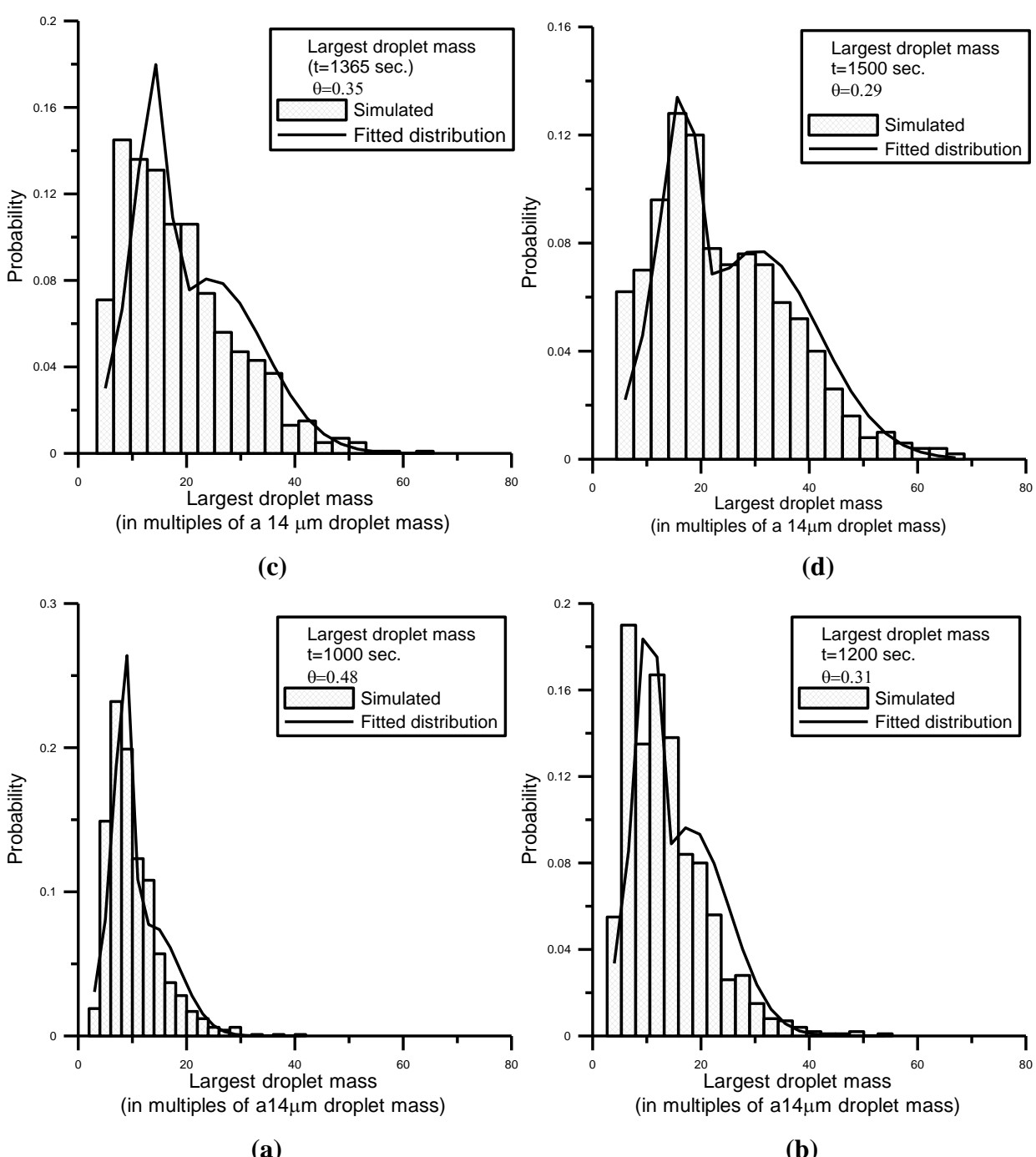


FIG. 1. (a)–(d) (dots) largest droplet mass distributions calculated from Monte Carlo simulations at four different times, for a system with an initial mono-disperse distribution of 100 droplets of 14 μm in radius; (solid line) fit using Eq. 6.


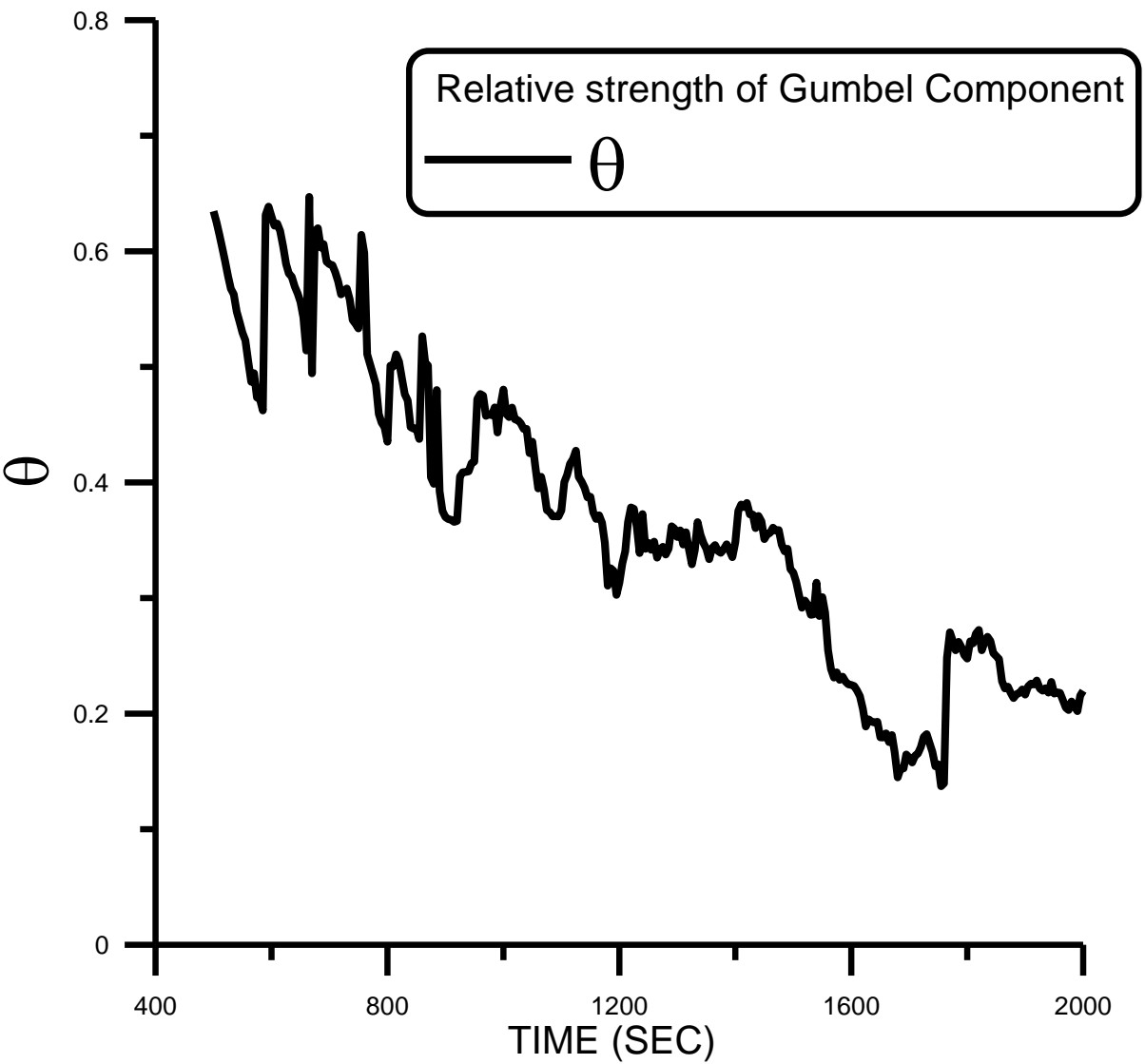

FIG. 2. Time evolution of the coefficient $\theta$ in Eq. 6, obtained for a simulation with the product kernel.


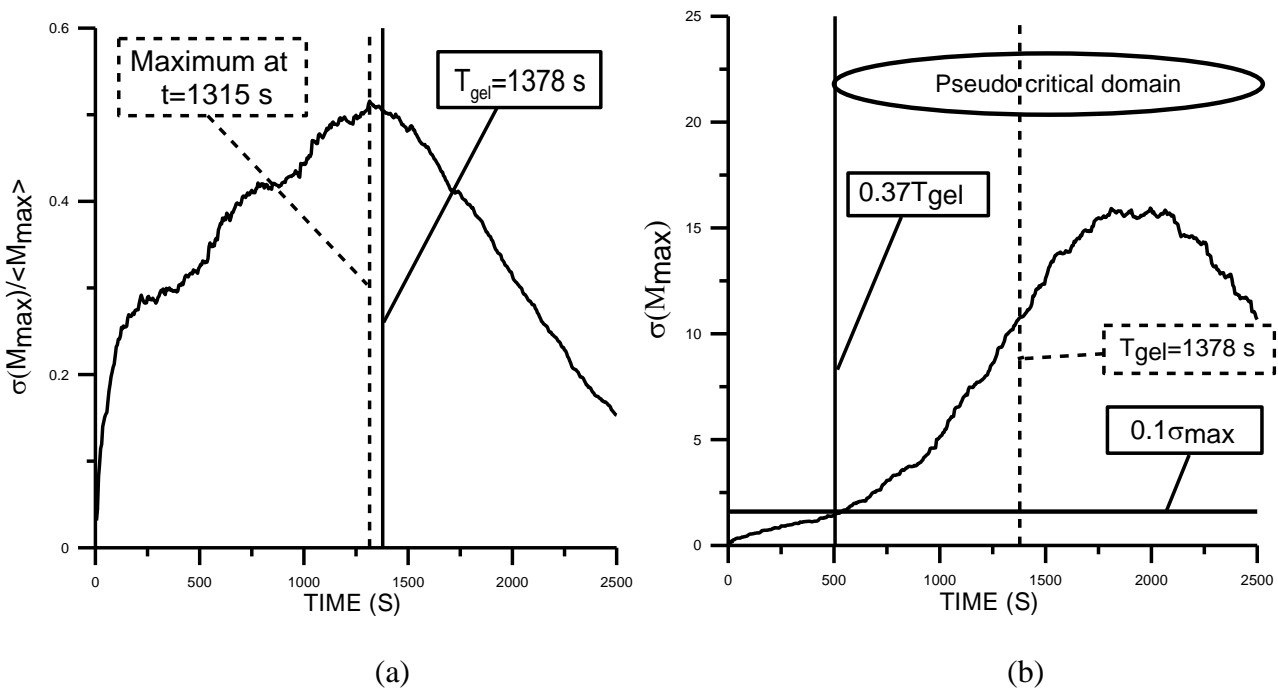

(a)                                    (b)

FIG 3. For the finite system, the normalized standard deviation $\sigma(M_{max})/\langle M_{max}\rangle$ of the largest droplet

mass versus time (Fig. 3a). The initial number of droplets was set equal to $N=100$ droplets of 14 μm

in radius in a volume of 1 cm$^3$. Simulations were performed with the product kernel $K(i,j) = Cx_i x_j$

(with $C= 5.49\text{x}10^{10}$ cm$^3$ g$^{-2}$ s$^{-1}$), and $N_r$=1000 realizations of the stochastic algorithm were performed.

The maximum value of $\sigma(M_{max})/\langle M_{max}\rangle$ is found to be 1315 sec. (dashed vertical line), and is very

close to the sol gel transition time (continuous vertical line) for the infinite system (1378 sec). In Fig.

3b the small end of the pseudo-critical domain is estimated as the time where $\sigma(M_{max}) = 0.1\sigma_{max}$ .


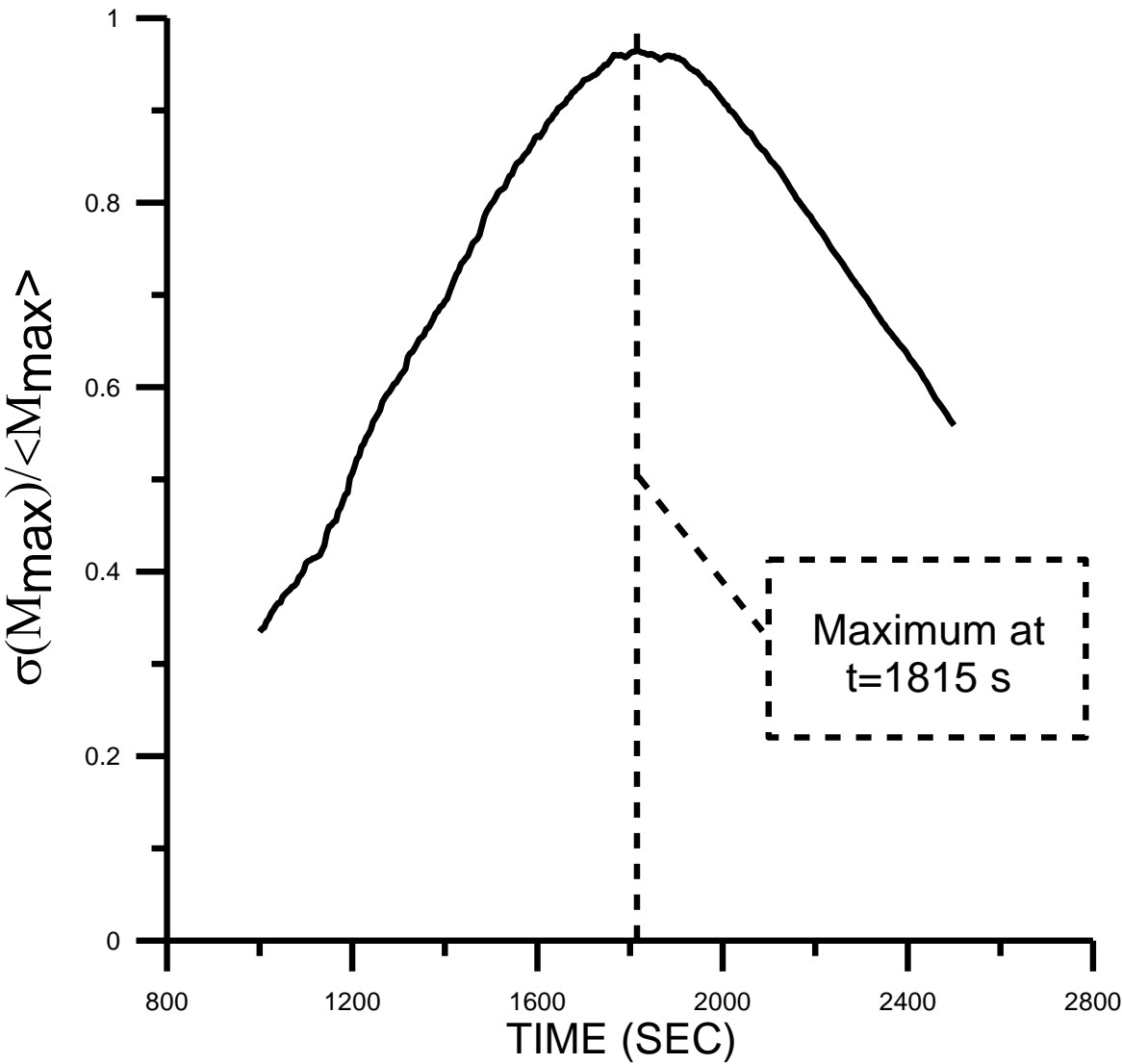

FIG. 4. Time evolution of the normalized standard deviation $\sigma(M_{max})/\langle M_{max}\rangle$ of the largest droplet

mass versus time estimated from the Monte Carlo algorithm. The simulations were performed for the

hydrodynamic kernel with a bidisperse initial condition (200 droplets of 10 μm in radius, and 50

droplets of 12.6 μm) in a volume of 1 cm$^3$.

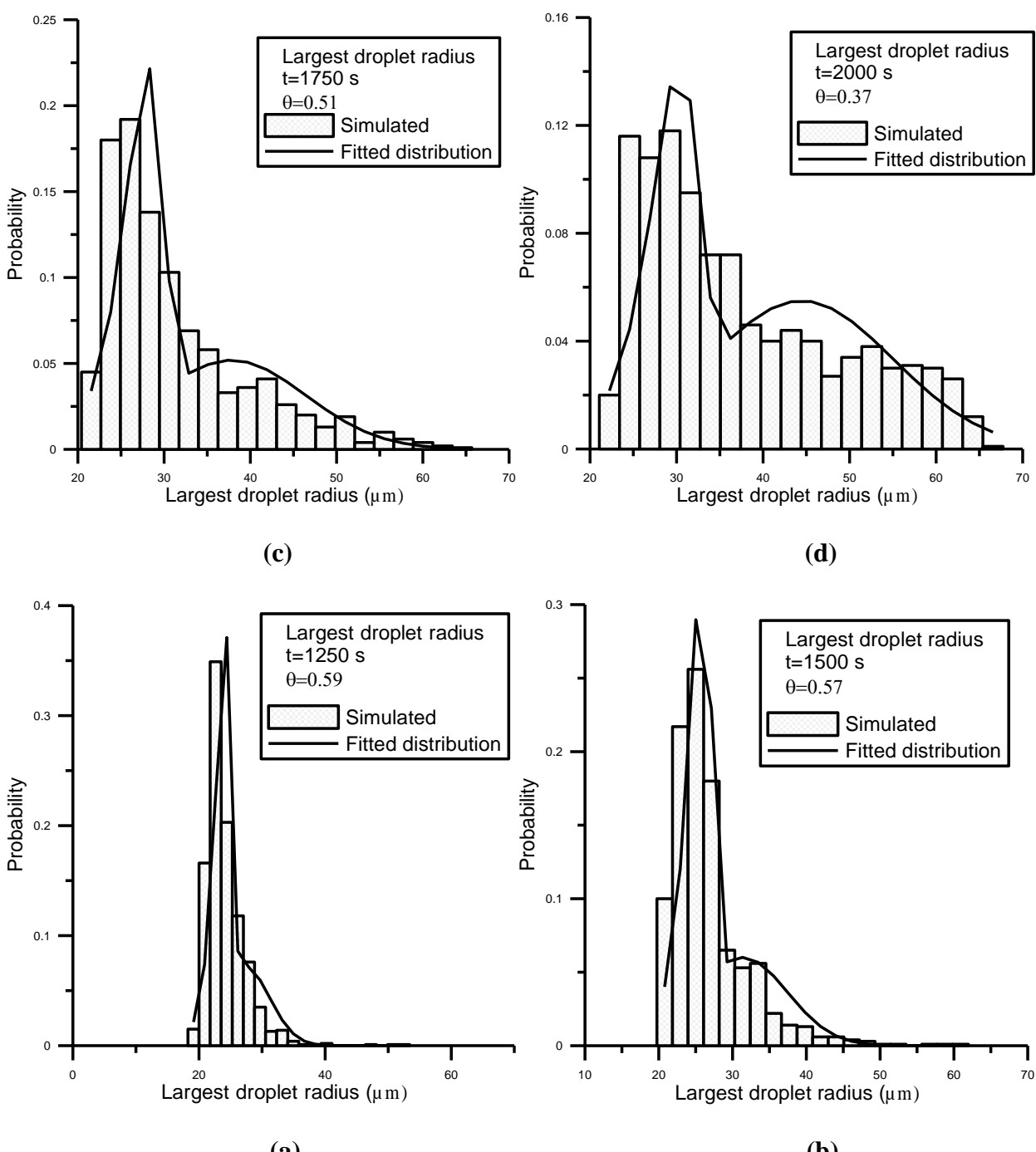


FIG. 5 (a)–(d) (dots) Simulated $M_{max}$ distributions in a system with an initial bidisperse distribution (200 droplets of 10 μm in radius, and 50 droplets of 12.6 μm) at four different times; (full line) fit using Eq. 6. The simulations were performed for the turbulent hydrodynamic kernel.


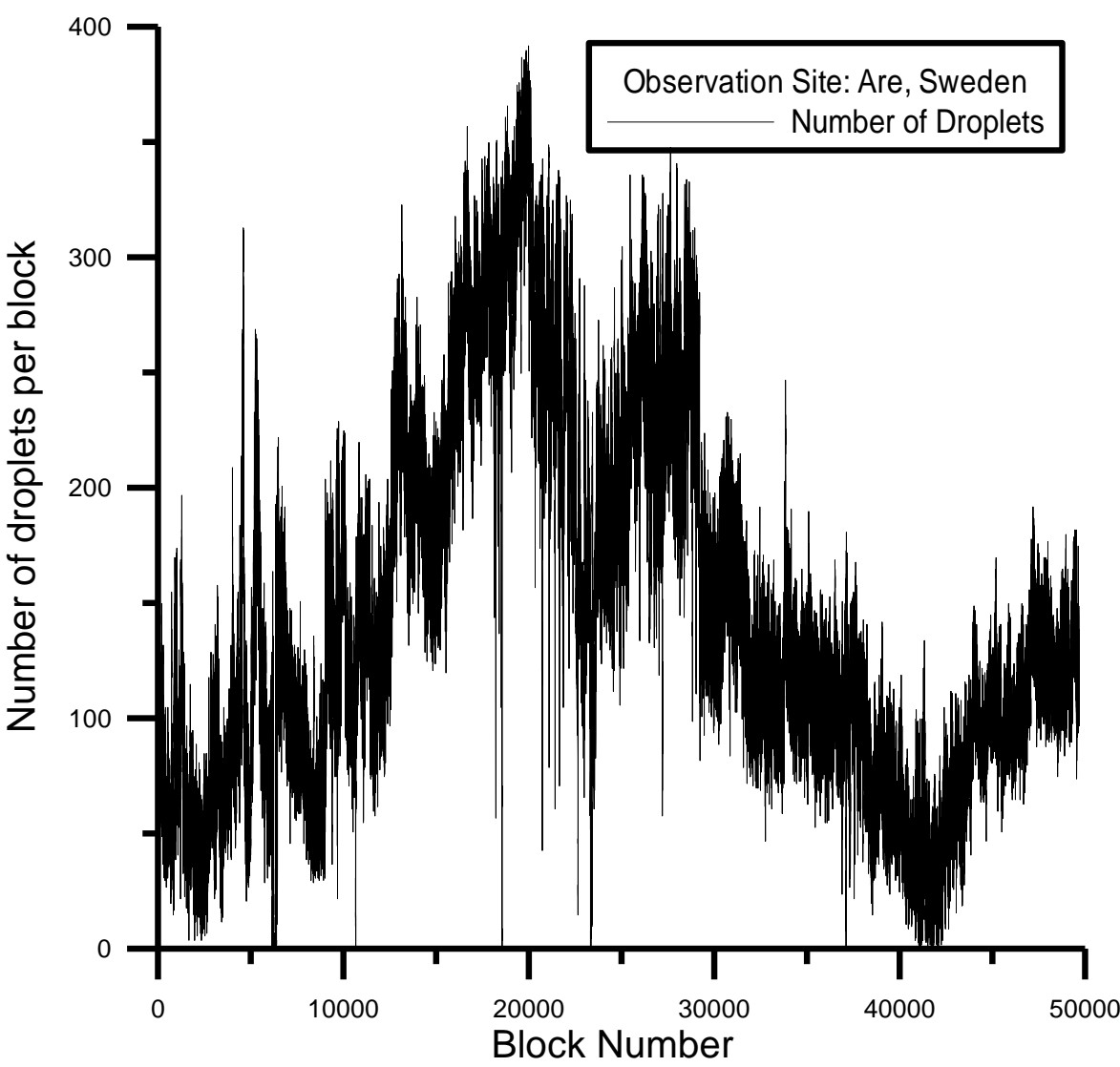

FIG. 6. Time series of the number of droplets per block, sampled at a hilltop in Are, Sweden.



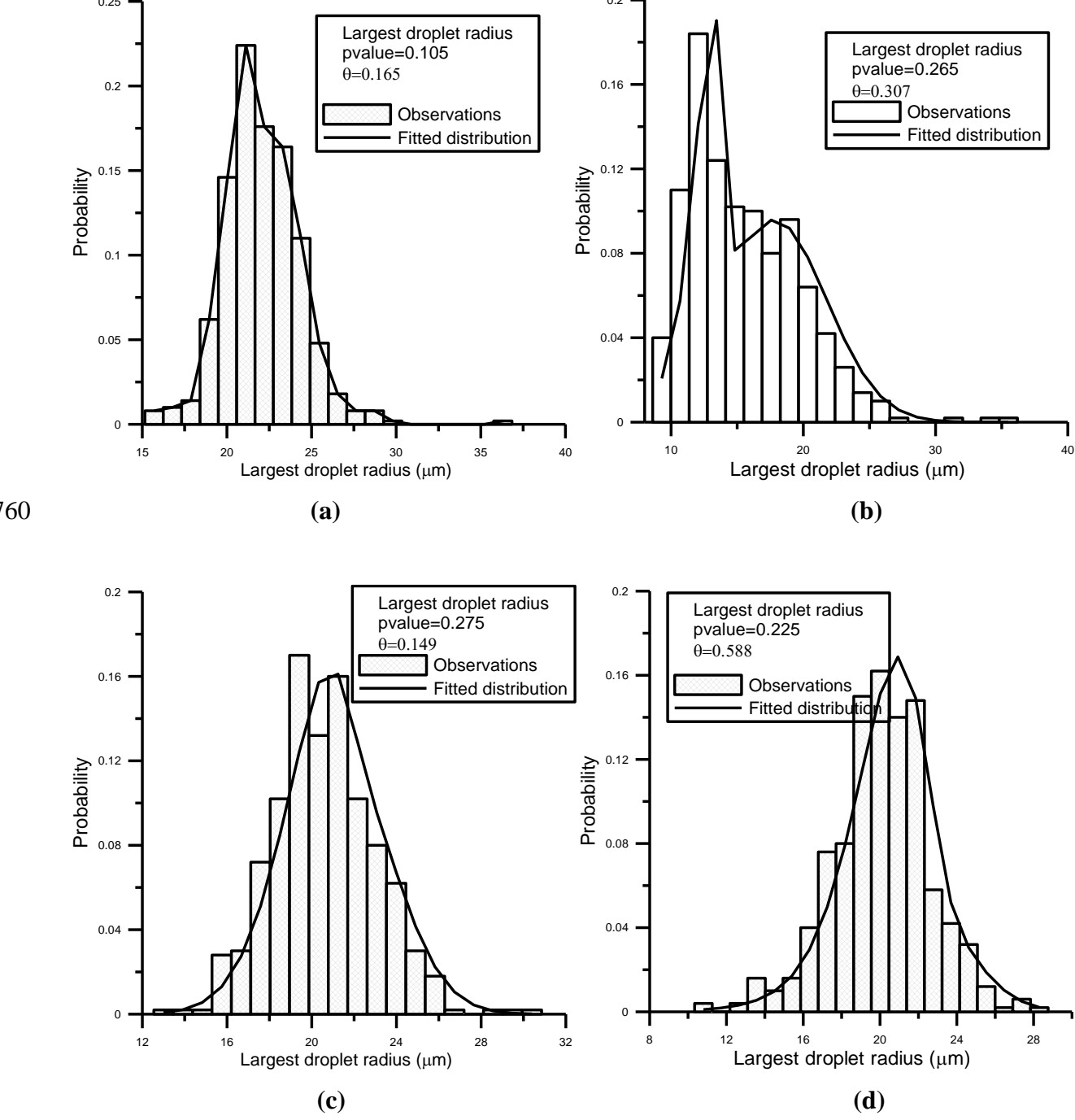

FIG. 7. For four random samples that are distributed following the admixture distribution (with sample

 size 500), observed (histogram) and fitted (solid line) using Eq. 6. Also shown for each distribution is the *p*-value of the goodness of fit test, and the parameter $\theta$ indicating the weight of the Gumbel component.

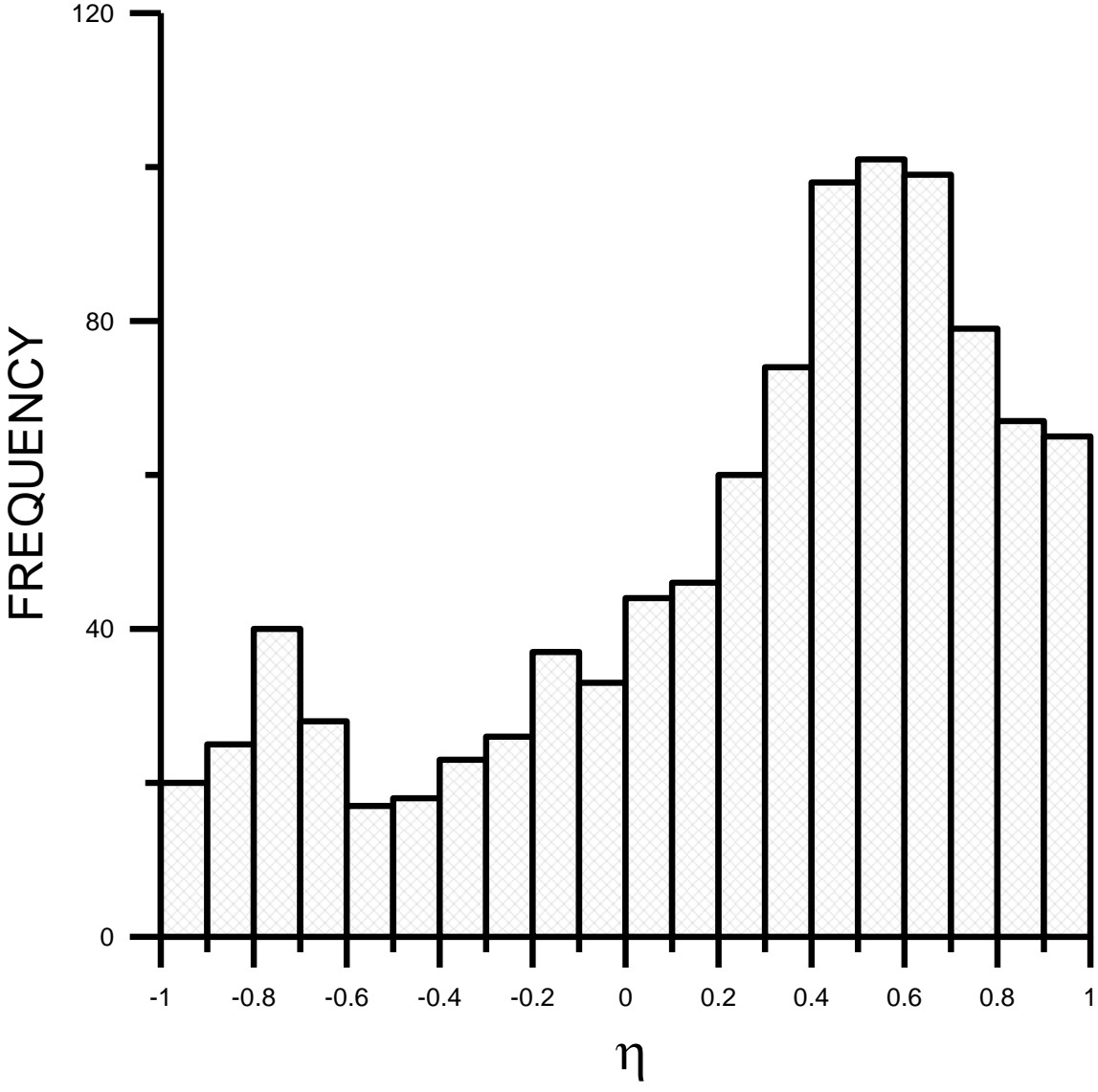

FIG. 8. Histogram of the ratio $\eta = (w_{Gaussian} - w_{Gumbel})/(w_{Gaussian} + w_{Gumbel})$, which measures the distance to the critical point.