# Peer review of "The impact of fluctuations and correlations in droplet growth by collision-coalescence revisited. Part II: Observational evidence of gel formation in warm clouds."

_Atmospheric Chemistry and Physics, 2018_

## Referee Comment (RC1) · Anonymous Referee #1 · 14 Feb 2019

General comments:

This manuscript is a sequel of "The impact of fluctuations and correlations in droplet growth by collision–coalescence revisited – Part 1: Numerical calculation of post-gel droplet size distribution" (https://doi.org/10.5194/acp-17-6895-2017). It aims to provide an observational evidence of "gel formation" in warm clouds. By analysing the droplet size distribution of the largest droplet from the observational data, the authors showed that the distribution of the mass of the largest droplet is a mixture of a Gaussian and a Gumbel distributions. In general, the idea and the corresponding analysis are original

in the sense that observational evidence was provided to support the "lucky droplet model". However, the authors didn't address the previous works on this topic. The state-of-art development on this topic is far beyond what is described in the current manuscript. I would support the publication of this manuscript if the following comments are carefully addressed.

Specific comments:

1. They are several studies that have already addressed the lucky droplet model for the collision-coalescence process of cloud droplets. I would suggest the authors cite those papers and address how the current manuscript advances the study compared with the earlier works. This can help place the current manuscript in a more general context and exhibit the novelty of the present study. Telford (1955) [1] may be the first to propose the lucky droplet model for the collision-coalescence process of cloud droplets. Kostinski and Shaw (2005) [2] developed the model of lucky droplet, which was further investigated using large deviation theory by Wilkinson (2016) [3]. The numerical work of Dziekan P, Pawlowska H. (2017) [4] supports the model of Kostinski and Shaw (2005) [2]. I suggest the authors to explicitly explain the main differences between the current study and those works mentioned above in the introduction and results.

2. Many papers have studied the the collision-coalescence problem, which should be also addressed in the introduction. A good summary is given by Grabowski and Wang (2013) [5]. The work (summarized in Grabowski and Wang (2013) [5]) by the group of Wang should be addressed. Several stochastic models by Pinsky et al (2004, 2007, 2008) [8]-[10], Mehlig et al (2007) [6], and Wilkinson et al (2006) [7] should be cited. Recent numerical work by Onishi and Seifert (2016) [11], Li et al (2017) [12], Li et al (2018) [13], and Chen et al. (2018) [14].

I would suggest the authors compare the Monte-Carlo method used in Shima et al. 2009 [15], Li et al (2017) [12] and Li et al (2018) [13].

3. For the fitted distribution in Fig.1, 5, and 7, could the authors have more samples to get better statistics?

4. A question related to question 3.: on Line 239, the authors used 200 droplets of 10um, and 50 droplets of 12.6 um for the Monte Carlo simulation. Is it statistically convergent? Can the authors provide a statistically convergent study (similar to the one in Li et al (2017) [12])?

5. L65: please provide reference for the use of "gel formation" in "percolation theory" and "nuclear physics" respectively.

6. L80: "average number of droplets". Do you mean "droplet (particle) number density" ? I would suggest the author use the commonly accepted terminology in both cloud physics and statistical mechanics for readability.

7. L81: I don't quite understand "the time rate of change of...". Could you please rephrase the sentence for readability?

8. Eq.3, where is "\tau" defined?

9. I don't understand how Eq.4 is obtained. What is $T\_gel$? What is the physics of this time scale?

10. For the discussion of the Smoluchowski equation in section 2.1, please compare the argument by Pumir, A., and M. Wilkinson, 2016 [16]. http://soft-matter.seas.harvard.edu/index.php/Sol-Gel_Transition

11. L110: please provide reference after "experimentally".

12. L111: please provide reference after "percolation".

13. Eq. 5a and 5b, please compare them with Kostinski and Shaw (2005) [2] and Wilkinson (2016) [3].

14. L133: "We must emphasize that phase transitions cannot take place in a finite system. For this type of systems, the notion of pseudo-critical region is introduced.". Please provide more physical explanation and references for the statement and "pseudo-critical region".

15. L154: What is "product kernel"? If it is widely used, please provide several references. What are the assumptions for the kernel, linear drag, gravity only? Could you please explain why you choose this kernel?

16. L164: Could you please explain what kinds of "Monte Carlo algorithm" you used? Is it comparable to Shima et al. 2009 [15], Li et al (2017) [12] and Li et al (2018) [13]? I understand you focus on the collision-coalescence process of cloud droplets. Could you please also provide the equations you solved numerically? Also, can you explain the difference of your "Monte Carlo algorithm" with those of Shima et al. 2009 [15], Li et al (2017) [12] and Li et al (2018) [13].

17. L167: Can you give a physical explanation about why you choose "C=5,49*10ˆ(10) cmˆ3sˆ{-1}"?

18. L173: Could you please explain more about the "mixing fraction", like mixing fraction of which quantity and the corresponding physical picture or intuition?

19. L250-259: Could you please describe in more details about the measurement, like where the cloud droplets are from, warm clouds? What are the measuring environmentally conditions, like the temperature, water vapor mixing ratio? What is the spatial and time resolution of the FM-120? Can you measure the time evolution of droplet size distribution?

20. L260: Please provide reference for "The block maxima (BM) approach in extreme value theory (EVT) was applied" and compare with the large deviation theory/method described in Wilkinson 16 [3].

Technical corrections:

21. L271-272: Please rephrase the sentence "The sample size...of data" to improve

the readability. The "which clause" is not encouraged in scientific writing.

22. L318: Did you mean "entire dataset"?

References

[1] Telford JW. 1955. A new aspect of coalescence theory. Journal of Meteorology 12(5): 436–444.

[2] Kostinski AB, Shaw RA. 2005. Fluctuations and luck in droplet growth by coalescence. Bull. Am. Met. Soc. 86: 235–244.

[3] Wilkinson M. 2016. Large deviation analysis of rapid onset of rain showers. Phys. Rev. Lett. 116: 018 501, doi:10.1103/PhysRevLett.116.018501, URL http://link.aps.org/doi/10.1103/PhysRevLett.116.018501.

[4] Dziekan P, Pawlowska H. 2017. Stochastic coalescence in lagrangian cloud microphysics. Atmospheric Chemistry and Physics 17(22): 13 509–13 520

[5] Grabowski, W. W., and L.-P. Wang, 2013: Growth of cloud droplets in a turbulent environment. Annu. Rev. Fluid Mech., 45 (1), 293–324.

[6] Mehlig, B., M. Wilkinson, and V. Uski, 2007: Colliding particles in highly turbulent flows. Phys. Fluids, 19, 098107.

[7] Wilkinson, M., B. Mehlig, and V. Bezuglyy, 2006: Caustic activation of rain showers. Phys. Rev. Lett., 97, 048 501.

[8] Pinsky, M., and A. Khain, 2004: Collisions of small drops in a turbulent flow. part ii: Effects of flow accelerations. J. Atmosph. Sci., 61 (15), 1926–1939.

[9] Pinsky, M., A. Khain, and H. Krugliak, 2008: Collisions of cloud droplets in a turbulent flow. part v: Application of detailed tables of turbulent collision rate enhancement to simulation of droplet spectra evolution. J. Atmosph. Sci., 65 (2), 357–374.

[10] Pinsky, M., A. Khain, and M. Shapiro, 2007: Collisions of cloud droplets in a
turbulent flow. part iv: Droplet hydrodynamic interaction. J. Atmosph. Sci., 64 (7), 2462–2482.

[11] Onishi, R., and A. Seifert, 2016: Reynolds-number dependence of turbulence enhancement on collision growth. Atmosph. Chemistry and Physics, 16 (19), 12 441–12 455.

[12] Li XY, Brandenburg A, Haugen NEL, Svensson G. 2017. Eulerian and l agrangian approaches to multidimensional condensation and collection. J. Adv. Modeling Earth Systems 9(2): 1116–1137.

[13] Li XY, Brandenburg A, Svensson G, Haugen NE, Mehlig B, Rogachevskii I. 2018b. Effect of turbulence on collisional growth of cloud droplets. Journal of the Atmospheric Sciences 75(10): 3469–3487.

[14] Chen, S., M. Yau, and P. Bartello, 2018: Turbulence effects of collision efficiency and broadening of droplet size distribution in cumulus clouds. J. Atmosph. Sci., 75 (1), 203–217.

[15] Shima, S., K. Kusano, A. Kawano, T. Sugiyama, and S. Kawahara, 2009: The super-droplet method for the numerical simulation of clouds and precipitation: a particle-based and probabilistic microphysics model coupled with a non-hydrostatic model. Quart. J. Roy. Met. Soc., 135, 1307–1320, physics/0701103.

[16] Pumir, A., and M. Wilkinson, 2016: Collisional aggregation due to turbulence. Annu. Rev. Condensed Matter Physics, 7, 141–170.

---

## Referee Comment (RC2) · Anonymous Referee #2 · 30 Aug 2019

The present manuscript is a follow-up article from the authors about cloud droplet collision-coalescence processes, which sought an answer to why the models take a longer time to form precipitation droplets than the observations suggests. Authors describe in their earlier article this is because of a few large droplets getting formed first, which then more efficiently collect smaller droplets in the collision-coalescence process and get bigger faster than others.

Authors perform an ensemble of simulations using the method developed by them and described in their earlier article. Then, they look into the frequency distribution of the

largest droplet radius across the number of simulations in the ensemble. Authors show that the frequency distribution is a sum of Gumbel and Gaussian distribution, and as the length of simulation time is increased, the weight of Gaussian distribution increases. Authors also show the frequency distribution of the largest droplet in a record of 4 hours of fog data is a sum of Gumbel and Gaussian distribution. The frequency distributions of maximum droplet size, as noticed in simulations and observations are held as evidence of the hypothesis mentioned in the beginning.

My primary contention is that the authors don't show why the said hypothesis (authors call it sol-gel hypothesis) will lead to a particular frequency distribution of the size of the biggest droplets or the shown size distribution is unique to only this hypothesis. Second, the connection between observed frequency distribution in fog data and simulations is tenuous. Third, sufficient details are not provided to ascertain whether the frequency distribution as the sum of two distributions with a particular value of weighting factor is better than say if only one of the distribution (that is weighting factor either zero or one) was fitted.

I am unconvinced that the manuscript presents observational evidence of gel formation in warm clouds.

---

## Author Response (AR1)

In this document, we provide a point by point response to all referees comments and specify all changes made in the revised manuscript. The additions are marked in red both in this document and in the revised manuscript. Section 1 is devoted to referee #1, section 2 is devoted to referee #2.

**Section 1: Response to Referee # 1 comments and modifications made to the manuscript (marked in red in the revised manuscript):**

**Anonymous referee # 1**

1. *They are several studies that have already addressed the lucky droplet model for the collision-coalescence process of cloud droplets. I would suggest the authors cite those papers and address how the current manuscript advances the study compared with the earlier works. This can help place the current manuscript in a more general context and exhibit the novelty of the present study. Telford (1955) [1] may be the first to propose the lucky droplet model for the collision-coalescence process of cloud droplets. Kostinski and Shaw (2005) [2] developed the model of lucky droplet, which was further investigated using large deviation theory by Wilkinson (2016) [3]. The numerical work of Dziekan P, Pawlowska H. (2017) [4] supports the model of Kostinski and Shaw (2005) [2]. I suggest the authors to explicitly explain the main differences between the current study and those works mentioned above in the introduction and results.*

**Reply to referee:**

In the revised version of our paper, we will expand the list of previous papers that addressed the lucky droplet model for the collision coalescence process (by including some papers suggested by the reviewer), and also will explicitly explain the main differences between previous studies and present work, both in the introduction and in the conclusions.

**On the main differences between previous studies and present work:**

Previous efforts on this direction were mainly focused on finding the distribution of times for *N* collisions (Telford, 1955; Kostinski and Shaw, 2005; Wilkinson; 2016), while we were concentrated on studying the "lucky droplet" size distribution to determine whether or not the runaway growth process due to collision-coalescence has started.

For example, Kostinski and Shaw (2005) present a distribution of the time to produce drizzle by calculating the convolution of the exponentially distributed times between collisions. They found the distribution of:

$$T_{N_c} = \sum_{i=1}^{N_c} t_i$$

(S1)

with $N_c$ fixed, where the $t_i$ are the times between droplet collisions and $N_c$ the number of collisions, which have an exponential distribution.

Wilkinson (2016) found the probability density for the time $T_{N_c} = \sum_{i=1}^{N_c} t_i$ being a small fraction of its average value (far from the mean value). As the precipitation occurs on a time scale that is smaller than the typical scale for one collision, the problem was solved by applying large deviation theory. **More details on large deviation theory (LDT) can be found in this document in the reply to question 20.**

**References:**

Kostinski, A. B., & Shaw, R. A. (2005). Fluctuations and luck in droplet growth by

      coalescence. *Bulletin of the American Meteorological Society*, *86*(2), 235-244.

Telford, J. W. (1955). A new aspect of coalescence theory. *Journal of Meteorology*, *12*(5),

      436-444.

Wilkinson M. 2016. Large deviation analysis of rapid onset of rain showers. Phys. Rev.

Lett. 116: 018 501, doi:10.1103/PhysRevLett.116.018501.

**Changes made in the revised manuscript:**

Now, in the introduction, the list of previous papers that addressed the lucky droplet model was expanded, and the differences with present work are highlighted. New references are added and the new additions to the manuscript are marked in red.

**Line 35 of the revised paper:**

Numerous mechanisms have been proposed to close the gap between observations and simulations. Some theories explain this phenomenon as an increase in collision efficiencies due to turbulence (Wang et al., 2008; Pinsky et al., 2004, 2007, 2008), turbulence-enhanced collision rate of cloud droplets (Falkovich and Pumir, 2007; Grabowski and Wang, 2013), and turbulent dispersion of cloud droplets (Sidin et al., 2009).

More recent papers (Onishi and Seifert, 2016; Li et al, 2017; Li et al, 2018, and Chen et al., 2018) also investigated the effect of turbulence in early development of precipitation.

**Line 70 of the revised paper:**

Telford (1955) may be the first to propose the "lucky droplet" model for collision-coalescence of cloud droplets. One of the novelties of Telford's approach was to recognize the shortcomings of the "continuous growth model", and took into account the statistical fluctuations inherent to the collision-coalescence process and its discrete nature. He performed his analysis for a cloud consisting of identical 10 µm droplets together with collector drops with twice the volume (12.6 µm radius). From this initial bimodal distribution, he found that 100 of the 12.6 µm droplets per cubic meter (a $10^{-6}$ fraction), will grow more rapidly than predicted by the continuous growth model, experiencing their first

10 coalescences after a time of approximately 5 minutes, while the time to undergo 10 collisions assuming continuous growth was about 33 minutes.

The lucky droplet model was further developed by Kostinski and Shaw (2005), who presented numerical evidence that their model can lead to a rapid development of precipitation. Their analysis was based on the derivation of the distribution of times for N collisions (which resulted to be the Erlang distribution). They concluded that the $10^{-6}$ lucky droplets are expected to reach the 50 µm 10 times faster than the average droplet. More recently, Wilkinson (2016) advanced further the model by using large deviation theory (Touchette, 2009). He derived the probability for the time T to undergo N collisions being a very small fraction of its mean value, and showed that the time scale for the initiation of precipitation is smaller than the mean time for a single collision.

The results obtained by Kostinski and Shaw (2005) were tested by Dziekan and Pawlowska (2017) by calculating the "luck factor", ie, how much faster the luckiest droplets grow to r=40 µm compared to the average droplets. They estimated that the luckiest $10^{-3}$ fraction will cross the size gap around 5 times faster, and the luckiest $10^{-5}$ fraction around 11 times faster, in good agreement with the results obtained by Kostinksi and Shaw (2005) (about 6 and 9 times faster respectively).

However, previous efforts on this direction were mainly focused on finding the distribution of times for *N* collisions (Telford, 1955; Kostinski and Shaw, 2005; Wilkinson; 2016), while we were concentrated on studying the "lucky droplet" size distribution to determine whether or not the runaway growth process due to collision-coalescence has started.

**Anonymous referee # 1**

*2. Many papers have studied the collision-coalescence problem, which should be also addressed in the introduction. A good summary is given by Grabowski and Wang (2013) [5]. The work (summarized in Grabowski and Wang (2013) [5]) by the group of Wang should be addressed. Several stochastic models by Pinsky et al (2004, 2007, 2008) [8]-[10], Mehlig et al (2007) [6], and Wilkinson et al (2006) [7] should be cited. Recent numerical work by Onishi and Seifert (2016) [11], Li et al (2017) [12], Li et al (2018) [13], and Chen et al. (2018) [14].*

**Reply to referee:**

As suggested by the reviewer, a list of papers that addressed the collision-coalescence problem will be added.

**Changes made in the revised manuscript:**

**A list of references were added, as suggested by the reviewer.**

References added:

Chen, S., Yau, M. K., and Bartello, P.: Turbulence effects of collision efficiency and broadening of droplet size distribution in cumulus clouds, J. Atmos. Sci., https://doi.org/10.1175/JAS-D17-0123.1, 2018.

Drake, R. L.: The scalar transport equation of coalescence theory: Moments and kernels, J. Atmos. Sci., 29, 537–547, 1972.

Drake, R.L. and Wright, T.J.: The scalar transport equation of coalescence theory: New families of exact solutions, J. Atmos. Sci., 29, 548-556, 1972.

Dziekan, P. and Pawlowska, H.: Stochastic coalescence in Lagrangian cloud microphysics, Atmos. Chem. Phys., 17, 13509– 13520, https://doi.org/10.5194/acp-17-13509-2017, 2017.

Falkovich, Gregory, and Alain Pumir.: Sling effect in collisions of water droplets in turbulent clouds. *Journal of the Atmospheric Sciences* 64(12), 4497-4505, 2007.

Golovin, A. M.: The Solution of the Coagulation Equation for Raindrops. Taking Condensation into Account. *Soviet Physics Doklady*. Vol. 8, 1963.

Johnson, D. B.: The role of giant and ultragiant aerosol particles in warm rain initiation. *Journal of the Atmospheric Sciences*, *39*(2), 448-460, 1982.

Kolb, M. and Axelos, A.V.: Gelation Transition versus Percolation Theory. *Correlations and Connectivity*. Springer, Dordrecht, 255-261, 1990.

Kostinski, A. B. and Shaw, R. A.: Fluctuations and luck in droplet growth by coalescence, B. Am. Meteorol. Soc., 86, 235–244, 2005.

Li, X.-Y., Brandenburg, A., Haugen, N. E. L., and Svensson, G.: Eulerian and Lagrangian approaches to multidimensional condensation and collection, J. Adv. Model. Earth Syst., 9, 1116–1137, https://doi.org/10.1002/2017MS000930, 2017.

Li, X.-Y., Brandenburg, A., Svensson, G., Haugen, N. E. L., Mehlig, B., and Rogachevskii, I.: Effect of Turbulence on Collisional Growth of Cloud Droplets, J. Atmos. Sci., 75, 3469–3487, https://doi.org/10.1175/JAS-D-18-0081.1, 2018.

Long, A.: Solutions to the droplet collection equation for polynomial kernels, J. Atmos. Sci., 31, 1040–1052, 1974.

Onishi, R. and Seifert, A.: Reynolds-number dependence of turbulence enhancement on collision growth, Atmos. Chem. Phys., 16, 12441–12455, https://doi.org/10.5194/acp-16-12441-2016, 2016.

Pinsky, M. B., and Khain, A. P.: Collisions of small drops in a turbulent flow. Part II: Effects of flow accelerations. *Journal of the atmospheric sciences*, *61*(15), 1926-1939, 2004.

Pinsky, M. B., A. P. Khain, and M. Shapiro.: Collisions of cloud droplets in a turbulent flow. Part IV: Droplet hydrodynamic interaction. *Journal of the atmospheric sciences* 64 (7), 2462-2482, 2007.

Scott, W.T.: Analytic studies of cloud droplet coalescence, J.Atmos. Sci., 25, 54-65, 1968.

Shima, S.-I., Kusano, K., Kawano, A., Sugiyama, T., and Kawahara, S.: The super-droplet method for the numerical simulation of clouds and precipitation: A particle-based and probabilistic microphysics model coupled with a non-hydrostatic model, Q. J. Roy. Meteorol. Soc., 135, 1307–1320, 2009.

Sidin, Ryan SR, Rob Hagmeijer, and Ulrich Sachs.: Evaluation of master equations for the droplet size distribution in condensing flow. *Physics of fluids* 21(7), 073303, 2009.

Telford, J.: A new aspect of coalescence theory, J. Meteorol., 12, 436–444, 1955.

Touchette, Hugo.: The large deviation approach to statistical mechanics. *Physics Reports,* 478.1-3, 1-69, 2009.

Van den Heever, S. C. and Cotton, W. R.: Urban aerosol impacts on downwind convective storms, J. Appl. Meteorol. Clim., 46, 828–850, 2007.

Wilkinson, M.: Large deviation analysis of rapid onset of rain showers. *Physical review letters* 116(1), 018501, 2016.

Yin, Y., Levin, Z., Reisin, T. G., and Tzivion, S.: The effect of giant cloud condensation nuclei on the  development of precipitation in convective clouds – A numerical study, Atmos. Res., 53, 91–116, 2000.

**Anonymous Referee # 1:**
  2. *I would suggest the authors compare the Monte-Carlo method used in Shima et al. 2009 [15], Li et al (2017) [12] and Li et al (2018) [13].*
**Reply to referee:**

**The Monte Carlo algorithm (Difference between the algorithm of Gillespie (1975) and the algorithm of Shima (2009)):**

In our study we use the stochastic simulation algorithm (SSA) developed by Gillespie (1976) for chemical reactions which rigorously account for fluctuations and correlations in a coalescing system. This algorithm was reformulated to simulate the kinetic behavior of aggregating systems by Laurenzi and Diamond (2002), by defining species as a type of aggregate with a specific size and composition. In our case, species represent droplets of different sizes.

The main difference between the Gillespie's SSA and the Monte Carlo method used in Shima et al. (2009), is that the SSA involved the collision of only two physical particles (droplets in our case) per MC cycle, while in the Super Droplet (SD) method developed by Shima et al. (2009) and other algorithms based on the simulation particles (SIP) approach (Li et al, 2017), in each MC cycle collide super-droplets, that represents a multiple number of droplets with the same attributes (radius $r$ or mass in the simplest case) and position.

For Gillespie's SSA the number of collisions ($C_T$) during a time interval $\Delta t$ can be estimated from the expression:

$$C_T = \frac{\Delta t \sum_{j=1}^{N_s} \sum_{i=1}^{N_s} K(i,j) N_i N_j}{V} \tag{S2}$$

In (1) $K(i,j)$ is the collection kernel, $V$ is the coalescence cell volume and $N_i$ are the number of particles in species with index $i$ (particles of the same radius).

As the number of collisions $C_T$ (see Eq. 1) will increase quadratically with the initial number of particles (Gillespie, 1975), we can conclude that the application of the SSA in systems involving a large number of particles, and with only two physical particles colliding per MC cycle is highly impractical. For example, in a three dimensional cloud model the typical coalescence cell has a volume of $10^9$ cm$^3$ and considering a droplet concentration at cloud base typical of maritime clouds ($10^2$ cm$^{-3}$), then the number of droplets will be about $10^{11}$. Then, in this case the Gillespie's SSA is not a suitable option due to the huge number of collisions in large volumes, and the high cost in computation.

The super-droplet method of Shima et al. (2009) was design to overcome this problem. The total collision rate in a time interval for this method can be calculated from the expression:

$$C_{T-SD} = \frac{\Delta t \sum_{j=1}^{N_{SD}} \sum_{i=1}^{N_{SD}} K(i,j) \max(\xi_i, \xi_j)}{V} \tag{S3}$$

where (2) $\xi_i$ is the super-droplet's multiplicity (number of physical droplets of the same radius) and $N_{SD}$ the number of super-droplets. From comparison of (S1) and (S2) it can be concluded that the number of super-droplet's collisions in a time interval it's much smaller. In the original paper Shima et al. (2009) compared the results obtained with Monte Carlo simulations with the SD method with numerical solutions of the kinetic collection equation

and confirmed that the SD method reproduces the solution of the KCE if the number of super droplets $N_{SD}$ is sufficiently large ( about $2^{17}$). However, Unterstrasser et al. (2017) show that convergence is possible with a number of super droplets in the order of $10^2$. Another simplification of the SD method is that instead of considering *N(N-1)/2* collision pairs only [*N/2*] non overlapping randomly selected pairs are considered.

As was stated before, the SSA of Gillespie (1975, 1976) rigorously account for fluctuations and correlations in a coalescing system, and the temporal evolution of mean values at each droplet size can be obtained by averaging over many runs. However, in order to obtain accurate solutions at the large end of the distribution, a large number of realizations is required. The alternative is the master equation (Bayewitz et al, 1974; Alfonso, 2015; Alfonso and Raga, 2017), which also accounts for fluctuations and correlations, and can serve as a reliable benchmark for different Monte Carlo methods.

Dziekan and Pawlowska (2017) performed "one to one" simulations (in that case the multiplicity is $\xi_i=1$, and *SD* is equivalent to a physical droplet) with the SD method of Shima (2009), compared the solutions with the master equation (Alfonso and Raga, 2017)   and found that both approaches are generally in agreement, only with some differences at the large end of the distribution. Simulations results by Unterstrasser (2018) also show a good correspondence with the master equation even at the large end of the droplet size distribution. Gillespie's (1975, 1975) SSA works perfect for our purposes because it rigorously account for fluctuations and correlations that are inherent to a finite system. **Due to the finiteness of the systems,** our simulations are performed for small volumes with small number of droplets (in the range 50-300 cm$^{-3}$).

**References:**

Alfonso, L.: An algorithm for the numerical solution of the multivariate master equation for stochastic coalescence. *Atmospheric Chemistry and Physics*, vol. 15, no 21, p. 12315-12326, 2015.

Alfonso, L. and Raga, G.B.: The impact of fluctuations and correlations in droplet growth by collision-coalescence revisited. Part I: Numerical calculation of post-gel droplet size distribution, *Atmos Chem. Phys.,* 17, 6895–6905, 2017.

Bayewitz, M.H., Yerushalmi, J., Katz, S., and Shinnar, R.: The extent of correlations in a stochastic coalescence process, *J. Atmos. Sci*., 31, 1604-1614, 1974.

Dziekan, P., & Pawlowska, H. (2017). Stochastic coalescence in Lagrangian cloud microphysics. *Atmospheric Chemistry and Physics*, *17*(22), 13509-13520.Gillespie, D.T.: An Exact Method for Numerically Simulating the Stochastic Coalescence Process in a Cloud, *J. Atmos. Sci*. 32, 1977-1989, 1975.

Gillespie, D.T.: An Exact Method for Numerically Simulating the Stochastic Coalescence Process in a Cloud, *J. Atmos. Sci*. 32, 1977-1989, 1975.

Gillespie, D. T. (1976). A general method for numerically simulating the stochastic time evolution of coupled chemical reactions. *Journal of computational physics*, *22*(4), 403-434.

Laurenzi, I. J., Bartels, J. D., and Diamond, S. L.: A general algorithm for exact simulation of multicomponent aggregation processes. *Journal of Computational Physics*, *177*(2), 418-449, 2002.

Li XY, Brandenburg A, Haugen NEL, Svensson G. 2017. Eulerian and l agrangian approaches to multidimensional condensation and collection. J. Adv. Modeling Earth Systems 9(2): 1116–1137.

Shima, S. I., Kusano, K., Kawano, A., Sugiyama, T., & Kawahara, S. (2009). The super-droplet method for the numerical simulation of clouds and precipitation: A particle-based and probabilistic microphysics model coupled with a non-hydrostatic model. *Quarterly Journal of the Royal Meteorological Society*, *135*(642), 1307-1320.

Unterstrasser, S., Hoffmann, F., & Lerch, M. (2017). Collection/aggregation algorithms in Lagrangian cloud microphysical models: rigorous evaluation in box model simulations. *Geoscientific Model Development*, *10*(4), 1521.

**Changes made in the revised manuscript:**

The difference between the Gillespie (1975) algorithm, and other Monte algorithms based on simulations particles, is now discussed in section 3.1

Line 205 of the revised paper: The main difference between the Gillespie's SSA and other Monte Carlo methods based on the simulation particles (SIP) approach (like the Super Droplet method developed by Shima et al., (2009)), is that the Gillespie's SSA involved the collision of only two physical particles (droplets in our case) per MC cycle, while in the approach based on SIP in each MC cycle collide SIP (super-droplets, for example) that represents a multiple number of droplets with the same attributes (radius *r* or mass in the simplest case) and position. However, Gillespie's SSA works perfectly for our purposes, because, due to the finiteness of our systems, our simulations are performed for small volumes with a small number of droplets (in the range 50-300 $cm^{-3}$).

**Anonymous Referee #1:**

*3. For the fitted distribution in Fig.1, 5, and 7, could the authors have more samples to get better statistics?*

**Reply:**

The number of realizations in our Monte Carlo algorithm, will be the sample size in the application of the Block Maxima approach (Figs 1 and 5 of the paper). That's why do not increase the number of realizations when generating synthetic data, taking into account that the number of realizations in the simulations (1000) must be close to the average number of blocks for which the largest droplet maxima can be fitted to the combined distribution (Eq. 6 of the paper).

On the other hand (analyzing the problem in terms of the accuracy needed for calculating the average values), the simulations were performed for 1000 realizations that is sufficiently to obtain the desired accuracy for the expected values:

$$\langle N \rangle = \frac{\sum_{i=1}^{N_r} N_i}{N_r}$$

(S4)

The errors of the stochastic procedure can be calculated following Gillespie (1975) from the expression:

$$\sigma(N(t)) = \left\{ \frac{1}{K}\sum_{i=1}^{K}\left[ N^i(t) \right]^2 - \left[ \frac{1}{K}\sum_{i=}^{K} N^i(t) \right]^2 \right\} = \langle N^2 \rangle - \langle N \rangle^2$$

(S5)

Where $\langle N \rangle$ the ensemble average and $N_i$ is the droplet concentration for each realization. The ensemble average will be estimated with the desired accuracy if the condition

$$\sigma(N(t))/\langle N(t) \rangle \ll 1$$

(S6)

fulfilled. The errors of the procedure can be checked in Fig. 5 of Alfonso et al. (2013), demonstrating that the Monte Carlo averages are calculated with the desired accuracy. For Fig. 7, the sample size (500) was set equal to the average number of blocks for which the largest droplet maxima can be fitted to the mixture of distributions.

**References:**

Alfonso, L., Raga, G. B., and Baumgardner, D.: The validity of the kinetic collection equation revisited–Part 3: Sol–gel transition under turbulent conditions. *Atmospheric Chemistry and Physics*, vol. 13, no 2, p. 521-529, 2013.

Gillespie, D.T.: An Exact Method for Numerically Simulating the Stochastic Coalescence Process in a Cloud, *J. Atmos. Sci*. 32, 1977-1989, 1975.

**Changes made in the revised manuscript:**

**This problem is now discussed in section 3.**

**Line 226 of the revised version:** The empirical distribution of the maxima was obtained for 1000 realizations of the stochastic algorithm. There is no need of a larger number of realizations to get a better statistics, since the number of realizations in our Monte Carlo algorithm must be equal to the sample size in the application of the block maxima (BM) approach (see the next section for more details). On the other hand, this number is not much bigger than the number of blocks in the data for which the largest droplet maxima was fitted to fog data.

**Anonymous Referee #1:**

**Reviewer:**
*4. A question related to question 3.: on Line 239, the authors used 200 droplets of 10um, and 50 droplets of 12.6 um for the Monte Carlo simulation. Is it statistically convergent? Can the authors provide a statistically convergent study (similar to the one in Li et al (2017) [12])?*
**Reply:**
The idea was to perform simulations for small systems (with a small number of particles) for which fluctuations and correlations are relevant. That's why the number of droplets per $cm^3$ use in the simulations are small, and of the same order of the droplet concentrations obtained from observations (which fluctuate between 0 and 392, with an average of 146). **This point will be clarified in the revised version of the paper.**

**Changes made in the revised manuscript:**
This question is now clarified in section 3.
**Line 303 of the revised manuscript:** As we want to perform simulations for small systems (with a small number of particles) for which fluctuations and correlations are relevant, the number of droplets per $cm^3$ use in the simulations are small. They are of the same order of the droplet concentrations for each block obtained from observations, which fluctuate between 0 and 392 $cm^{-3}$, with an average of 146 $cm^{-3}$(see Fig.6).

**Reviewer:**
*5. L65: please provide reference for the use of "gel formation" in "percolation theory" and "nuclear physics" respectively.*
**Reply:**
The corresponding references will be added in the revised version:

Botet, R., Płoszajczak, M., Chbihi, A., Borderie, B., Durand, D.,and Frankland, J.: Universal fluctuations in heavy-ion collisions in the Fermi energy domain. *Physical Review Letters*, *86*(16), 3514, 2001
Botet, R. and Płoszajczak, M.: Exact order-parameter distribution for critical mean-field percolation and critical aggregation. *Physical Review Letters*, *95*(18), 185702, 2005.
Gruyer, D., Frankland, J. D., Botet, R., Płoszajczak, M., Bonnet, E., Chbihi, A., ... and Guinet, D.: Nuclear multifragmentation time scale and fluctuations of the largest fragment size. *Physical review letters*, *110*(17), 172701, 2013.

**Changes made in the revised manuscript:**

The corresponding references were added (new additions are marked in red).

**Line 97 of the revised version:** The aim of the present work here is to find observational evidence of gel formation, taking as a reference recent studies in percolation theory (Botet and Płoszajczak, 2005) and nuclear physics (Botet et al., 2001; Gruyer et al., 2013), which can shed some light on the gel (largest droplet) size distribution during the initial stage of precipitation formation.

**Anonymous referee #1:**

*6. L80: "average number of droplets". Do you mean "droplet (particle) number density" ? I would suggest the author use the commonly accepted terminology in both cloud physics and statistical mechanics for readability.*

**Reply:**

As was stated by Gillespie (1972), the definition of $N(i,t)$ in the kinetic collection equation (Eq. 1 of the paper), vary from author to author, but usually is taken to be **the average concentration of cloud droplets of mass *i* at time *t*.** In the revised version we will clarify this point in order to avoid confusions, as suggested.

**Changes made in the revised manuscript:**

The corresponding references were added (new additions are marked in red).

**Line 115 of the revised version:** where $N(i,t)$ is the average concentration of droplets with mass $x_i$ at time t, and $K(i,j)$ is the coagulation kernel related to the probability of coalescence of two drops of masses $x_i$ and $x_j$.

**Anonymous referee #1:**

*7. L81: I don't quite understand "the time rate of change of...". Could you please rephrase the sentence for readability?*

**Reply to referee:**

We will rewrite the sentence in the revised version as suggested.

**Changes made in the revised manuscript:**

The sentence was deleted to avoid confusions.

*Anonymous referee #1:*

*8. Eq.3, where is "\tau" defined?*

**Reply to referee:**

There is a mistake in Eq.3, we should have written $t$ instead of $\tau$.

**Changes made in the revised manuscript:**

The equation is now in its correct form.

Line 125 of the revised manuscript: $M_2(t) = \dfrac{M_2(t_0)}{1 - CM_2(t_0)t}$ $\qquad\qquad$ (3)

*Anonymous referee #1:*

*9. I don't understand how Eq.4 is obtained. What is T_gel? What is the physics of this time scale?*

**Reply:**

$T_{gel}$ is defined as the time when the second moment $M_2(t) = \dfrac{M_2(t_0)}{1 - CM_2(t_0)t}$ becomes infinite,

then $1 - CM_2(t_0)t = 0$, and $T_{gel} = \left[CM_2(t_0)\right]^{-1}$. The equation for $M_2(t)$ (moment of order 2) can be obtained from the general equation for moment evolution that was obtained by Drake (1972) from the continuous form of the KCE. It has the form:

$$\frac{dM_n(t)}{dt} = \frac{1}{2}\int_0^\infty\int_0^\infty \left[(x+y)^n - x^n - y^n\right]K(x,y)N(x,t)N(y,t)dxdy \qquad (S7)$$

In (7), $K(x,y)$ is the collection kernel $N(x,t)$ is the average droplet concentration. If we consider the product kernel $K(x,y) = C(xy)$ in equation (7), then, the equation for the second moment is:

$$\frac{dM_2(t)}{dt} = C[M_2(t)]^2 \qquad (S8)$$

So that $M_2(t) = \dfrac{M_2(t_0)}{1 - CM_2(t_0)t}$

After $T_{gel}$, the total, a runaway droplet forms, and the kinetic collection equation is no longer valid, since the assumption of a continuous distribution breaks down. There is in essence a

phase transition in the system from a continuous distribution to a continuous distribution *plus* a runaway droplet.

**References:**

Drake, R.L.: The scalar transport equation of coalescence theory: Moments and kernels, J. Atmos. Sci., 29, 537-547, 1972.

**Changes made in the revised manuscript:**

An appendix was added.

**Line 447:**

**1. Appendix A.**

The sol-gel transition time $T_{gel}$ is defined as the time when the second moment

$M_2(t) = \dfrac{M_2(t_0)}{1 - CM_2(t_0)t}$ becomes infinite, then $1 - CM_2(t_0)t = 0$, and $T_{gel} = \left[CM_2(t_0)\right]^{-1}$. The

equation for $M_2(t)$ (moment of order 2 with respect to mass) can be found from the general

equation for moment evolution that was obtained by Drake (1972) from the continuous form

of the kinetic collection equation (1). It has the form:

$$\frac{dM_n(t)}{dt} = \frac{1}{2}\int_0^\infty \int_0^\infty \left[(x+y)^n - x^n - y^n\right]K(x,y)N(x,t)N(y,t)dxdy \qquad (A1)$$

In (7), $K(x,y)$ is the collection kernel, $N(x,t)$ is the average droplet concentration and $x$ is the

droplet mass. If we consider the product kernel $K(x,y) = C(xy)$ in (A1), then, the equation

for the second moment is:

$$\frac{dM_2(t)}{dt} = C[M_2(t)]^2 \qquad (A2)$$

With solution $M_2(t) = \dfrac{M_2(t_0)}{1 - CM_2(t_0)t}$

After $T_{gel}$, a runaway droplet forms, and the kinetic collection equation is no longer valid, since the assumption of a continuous distribution breaks down. There is in essence a phase transition in the system from a continuous distribution to a continuous distribution *plus* a runaway droplet.

**Anonymous Referee # 1:**
*11. L110: please provide reference after "experimentally".*
**Reply to referee:**
The corresponding reference was added:

Lushnikov, A. A., Negin, A. E., & Pakhomov, A. V. (1990). Experimental observation of the aerosol-aerogel transition. *Chemical physics letters*, *175*(1-2), 138-142**.**

**Changes made in the revised manuscript:**

The corresponding reference was added:

**Line 145 of the revised manuscript:** The sol-gel transition has been observed experimentally, for example: aerogels in aerosol physics (Lushnikov et al., 1990),

**Anonymous Referee # 1:**
*12. L111: please provide reference after "percolation".*
**Reply to referee:**
The corresponding reference will be added added:

Kolb, M., & Axelos, M. A. (1990). Gelation Transition versus Percolation Theory. In *Correlations and Connectivity* (pp. 255-261). Springer, Dordrecht.

**Changes made in the revised manuscript:**

**The corresponding reference was added.**

**Line 146 of the revised manuscript:** and in other theoretical models such as that of percolation (Botet and Płoszajczak, 2005; Kolb and Axelos, M. A., 1990) where there is a close analogy between percolation and droplet coagulation.

**Anonymous referee #1:**
*13. Eq. 5a and 5b, please compare them with Kostinski and Shaw (2005) [2] and Wilkinson (2016) [3].*
**Reply to referee:**
Our approach is different from that of the mentioned authors. The Kolmogorov distribution (5a, 5b) and the mixture of a Gaussian and a Gumbel (6) are the distributions of the largest cluster (droplet) mass at critical point (Eqs, 5a, 5b) and in the pseudocritical region (Eq. 6) respectively. Kostinski and Shaw (2005) and Wilkinson (2016) were interested in the distribution **of times for $N$ collisions.**

**Changes made in the revised manuscript:**

This problem was now briefly discussed in the introduction.

Line 92 of the revised version: However, previous efforts on this direction were mainly focused on finding the distribution of times for $N$ collisions (Telford, 1955; Kostinski and Shaw, 2005; Wilkinson; 2016), while we were concentrated on studying the "lucky droplet" size distribution to determine whether or not the runaway growth process due to collision-coalescence has started.

**Anonymous Referee #1:**
*14. L133: "We must emphasize that phase transitions cannot take place in a finite system. For this type of systems, the notion of pseudo-critical region is introduced".Please provide more physical explanation and references for the statement and "pseudo-critical region".*
**Reply to referee:**
In theory of critical phenomena, a phase transition is defined as a singularity in the free-energy or any thermodynamic property of a system, which is proportional to the logarithm of the sum of exponentials. For finite-sized systems, the free energy is proportional to the logarithm of a finite number of exponentials, which are always positive. Then, such singularities are only possible within infinite systems by taking the thermodynamic limit $N \to \infty$ (Bhattacharjee, 2001). For example, for the the Smoluchowski model (which is

obtained in the thermodynamic limit), there is a disordered phase (before the sol gel transition), and ordered phase (after the sol gel transition), and $T_{gel} = \left[CM_2(t_0)\right]^{-1}$ is the critical point for the infinite system. Then, a phase transition cannot take place in a finite system.

For an infinite system, fluctuations and correlations are neglected, and become important as the system approaches the critical point, where the correlation length diverges and there is power-law divergence of some quantities (for example, for the Smoluchowski model $M_2(t) = \dfrac{M_2(t_0)}{1 - CM_2(t_0)t}$ ).

With decreasing size of the system, fluctuations and correlations become more important (Gruyer et al, 2013). There is no divergence of the second moment $M_2(t)$ (because such singularities are only possible within infinite systems), but it is expected to reach a maximum for a time close $T_{gel} = \left[CM_2(t_0)\right]^{-1}$. For this kind of systems there is an entire region with large fluctuations (in the vicinity of $T_{gel} = \left[CM_2(t_0)\right]^{-1}$): the "pseudo critical region".

**References:**

Bhattacharjee, S. M. (2001). Critical Phenomena: An Introduction from a modern perspective. In *Field Theories in Condensed Matter Physics* (pp. 69-117). Hindustan Book Agency, Gurgaon.

**Changes made in the revised manuscript:**

This problem is now discussed in more detail. On the other hand, the notion of "pseudocritical region" was analyzed in the "Conclusion and Discussion" section of the original manuscript.

**Line 170 of the revised manuscript:** We must emphasize that phase transitions cannot take place in a finite system. This is due to the fact that a phase transition is defined as a singularity in the free energy or any thermodynamic property of a system; and for finite-sized systems, the free energy is proportional to the logarithm of a finite number of exponentials, which are always positive. Consequently, those singularities are only possible within infinite systems by taking the thermodynamic limit. Then, for finite systems, the notion of pseudo-critical region is introduced (which is the finite system equivalent of a sol-gel transition time).

**The notion of pseudocritical region was already discussed in the original version of the paper ("Discussion and Conclusions"), line 417:** In the pseudo-critical phase, the fluctuations and correlations are no longer negligible and the distribution is of neither one nor the other asymptotic forms (Gumbel or Gaussian). In this case, the fit of the largest droplet mass (gel), is a mixture of a Gumbel (disordered state) and Gaussian (ordered state) distributions. As was demonstrated in the preceding section, this combined distribution (Eq. 6) is a good approximation to the largest droplet distribution (gel) in the pseudo-critical region. The fact that the mixture of distributions provides a better fit than the Gumbel and Gaussian distributions shows that the samples selected in our study are mainly in the pseudo-critical phase. To confirm this fact, the ratio $\eta$ was calculated for 1000 samples of size $n=500$ selected randomly from the data. Figure 8 shows that for 90% of the samples the ratio $\eta$ lies in the interval [-0.9, 0.9], clearly indicating that samples are in the pseudo-critical region.

**Reply to anonymous referee #1:**
*15. L154: What is "product kernel"? If it is widely used, please provide several references. What are the assumptions for the kernel, linear drag, gravity only? Could you please explain why you choose this kernel?*
**Reply to referee:**
The product kernel is a kernel proportional to the product of the masses of the colliding particles $K(i,j) = Cx_i x_j$. It is widely used because analytical solutions of the kinetic collection (KCE) or Smoluchowski equation (Eq. 1) have been obtained tor this kernel Golovin (1963), Scott (1968), Drake (1972) and Drake and Wright (1972). Additionally, analytical solutions have also been obtained for the constant and sum (with probability of collision proportional to the sum of the masses of the colliding particles) kernels.
Lushnikov (1978, 2004) and Tanaka and Nakazawa (1993) also obtained analytical solutions of the master equation for the product kernel. Also, it is widely known that the product kernel is a gelling kernel (Lushnikov, 1978; 2004), and Lushnikov (1978, 2004) analytically obtained post gel particle size distributions for this case.
The aforementioned factors explain why we chose this kernel: It is a gelling kernel, and (due to the existence of analytical solutions), there is analytical expression for the sol gel transition time ($T_{gel} = \left[ CM_2(t_0) \right]^{-1}$ ) for this case. Then, it served as a benchmark for our Monte Carlo experiments, and to evaluate our method for the calculation of the pseudo-critical region.

The value of the constant $C$ ($C=5.49\times10^{10}$) in the product kernel $K(i,j)=Cx_ix_j$ is a result of the polynomial approximation (Long, 1974):

$$K(x,y)=A+B(x+y)+Cxy \tag{S9}$$

of the hydrodynamic collection kernel:

$$K(x,y)=\pi\left[R(x)+r(y)\right]^2 E(x,y)\left[V(x)-V(y)\right] \tag{S10}$$

Long (1974) calculated the coefficients for the polynomials (9) approximating the hydrodynamic kernel (10) when the largest of the colliding drops is smaller than 50 µm. The results obtained by Long (1974) are displayed in the table S1 (Alfonso et al, 2008).

**Table S1. Polynomials approximating the actual collection kernel _K(x,y)_ (Long, 1974).**

| Approximating Polynomial _P(x,y)_ | Coefficients $R\le 50\,\mu\mathrm{m}$ (cm$^3$ sec$^{-1}$) |
|---|---|
| $K(x,y)=A$ | A=$1.20\times10^{-4}$ |
| $K(x,y)=A+B(x+y)$ | A=0

 B=$8.83\times10^{2}$ |
| $K(x,y)=Cxy$ | C=$5.49\times10^{10}$ |
| $K(x,y)=A+B(x+y)+Cxy$

 $A=B^2/C$ | A=$4.41\times10^{-7}$

 B=$1.36\times10^{2}$

 C=$4.18\times10^{10}$ |
| $K(x,y)=A+B(x+y)+Cxy$ | A=0

 B=$4.16\times10^{2}$

 C=$2.24\times10^{10}$ |

**References:**
Drake, R.L.: The scalar transport equation of coalescence theory: Moments and kernels, J. Atmos. Sci., 29, 537-547, 1972.
Drake, R.L., Wright, T.J.: The scalar transport equation of coalescence theory: New families of exact solutions, J. Atmos. Sci., 29, 548-556, 1972.
Golovin, A.M.: The solution of the coagulating equation for cloud droplets is a rising air current, Bull. Acad. Sci. USSR, Geophys. Ser., No. 5, 482-487, 1963.

Long, A. B. (1974). Solutions to the droplet collection equation for polynomial kernels. *Journal of the Atmospheric Sciences*, *31*(4), 1040-1052.

Lushnikov, A. A.: Coagulation in finite systems. *Journal of Colloid and Interface Science*, vol. 65, no 2, p. 276- 285, 1978.

Lushnikov, A. A.: From sol to gel exactly. *Physical review letters*, vol. 93, no 19, p. 198302, 2004.

Scott, W.T.: Analytic studies of cloud droplet coalescence, J.Atmos. Sci., 25, 54-65, 1968.

Tanaka, H., Nakazawa, K.: Stochastic coagulation equation and the validity of the statistical coagulation equation, *J. Geomag. Geoelecr*., 45, 361-381, 1993.

**Changes made in the revised manuscript:**

The product kernel was discussed, and some new references were added.

**Line 221 of the revised version:** The product kernel is proportional to the product of the masses of the colliding droplets. It is widely used because analytical solutions of the KCE or Smoluchowski equation (Eq. 1) have been obtained tor this kernel Golovin (1963), Scott (1968), Drake (1972) and Drake and Wright (1972). The value of the constant C ($C=5.49\times10^{10}$) in the product kernel is the result of the polynomial approximation $K(x, y) = A + B(x + y) + Cxy$ (Long, 1974) of the hydrodynamic collection kernel (Eq. 11).

**Anonymous Referee #1:**
*16. L164: Could you please explain what kinds of "Monte Carlo algorithm" you used?*
*Is it comparable to Shima et al. 2009 [15], Li et al (2017) [12] and Li et al (2018) [13]?*
*I understand you focus on the collision-coalescence process of cloud droplets. Could*
*you please also provide the equations you solved numerically? Also, can you explain*
*the difference of your "Monte Carlo algorithm" with those of Shima et al. 2009 [15], Li et al*
*(2017) [12] and Li et al (2018) [13].*
**Reply to referee:**
In this study we use the stochastic algorithm developed by Gillespie (1976) for chemical reactions. This algorithm was reformulated to simulate the kinetic behaviour of aggregating systems by Laurenzi and Diamond (1999), by defining species as a type of aggregate with a specific size and composition. In our case, species represent droplets of different sizes.
**As was remarked in the reply to question (2)**, the main difference between the Gillespie's stochastic simulation algorithm (SSA) and other Monte Carlo methods based on the simulation particles (SIP) approach (like the Super Droplet method developed by Shima et al. (2009)), is that the Gillespie's SSA involved the collision of only two physical particles (droplets in our case) per MC cycle, while in the approach based on SIP in each MC cycle collide SIP (super-droplets, for example) that represents a multiple number of droplets with the same attributes (radius *r* or mass in the simplest case) and position.

*Could you please also provide the equations you solved numerically?*

The **Gillespie (1976) algorithm** generates a statistically correct trajectory (possible solution) of the master equation. The steps below summarize the algorithm:

1) **Initialization:** Initialize the number of droplets in each species (the species are defined as droplets of different sizes). There is a unique index $\mu$ for each pair of droplets *i, j* that may collide. For a system with $N$ species $\left( n_1, \quad n_{2,} \quad ... \quad , n_N \right)$ $\mu \in \dfrac{N(N+1)}{2}$. The set $\{\mu\}$ defines the total collision space, and is equal to the total number of possible interactions.

2) **Monte Carlo step:** Generate random numbers to determine the next collision to occur, as well as the time to the next collision. The next collision $\mu$ is calculated according to the distribution $P_2(\mu) = \dfrac{a_\mu}{\alpha}$, where $a_\mu$ are calculated from the probabilities:

$$a(i,j) = V^{-1} K(i,j) n_i n_j dt \equiv \Pr\{ \text{ Probability that two unlike particles } i \text{ and } j$$
with populations (number of particles) $n_i$ and $n_j$ will collide within the imminent time interval

$$a(i,i) = V^{-1} K(i,i) \frac{n_i(n_i - 1)}{2} dt \equiv \Pr\{ \text{ Probability that two particles of the same}$$
species *i* with population (number of particles) $n_i$ collide within the imminent time interval}

and $\alpha = \sum\limits_{v=1}^{\frac{N(N+1)}{2}} a_v$. The time to the next collision is exponentially distributed with mean $1/\alpha$

3) **Update**: Increase the time by the randomly generated time in Step 2. Update the droplet count based on the collision that occurred.
4) **Iterate**: Go back to Step 2 unless the number of droplets is zero or the simulation time has been exceeded.

In the revised version, the details of the Monte Carlo method will be discussed in one appendix.

**Changes made in the revised manuscript:**

The main differences between the Gillespie Monte Carlo algorithm and other Monte Carlo methods based on simulation particles were discussed in section 3.1. On the other hand, an appendix describing the Monte Carlo algorithm was added:

**Line 200 of the revised manuscript:** For synthetic data analysis, the empirical distributions of the largest droplet mass ($M_{max}$) were obtained from Monte Carlo simulations, following Botet (2011). The species accounting formulation (Laurenzi et al., 2002) of the stochastic simulation algorithm (SSA) of Gillespie (1975) that rigorously accounts for fluctuations and correlations in a coalescing system was used for the stochastic simulation in this work (See Appendix B).

The main difference between the Gillespie's SSA and other Monte Carlo methods based on the simulation particles (SIP) approach (like the Super Droplet method developed by Shima et al., (2009)), is that the Gillespie's SSA involved the collision of only two physical particles (droplets in our case) per MC cycle, while in the approach based on SIP in each MC cycle collide SIP (super-droplets, for example) that represents a multiple number of droplets with the same attributes (radius $r$ or mass in the simplest case) and position. However, Gillespie's SSA works perfectly for our purposes, because, due to the finiteness of our systems, our simulations are performed for small volumes with a small number of droplets (in the range 50-300 $cm^{-3}$).

**A new appendix describing the Monte Carlo algorithm was added:**

Line 463 of the revised version:

**Appendix B: The Monte Carlo algorithm.**

In this study, the species accounting formulation (Laurenzi et al., 2002) of the stochastic simulation algorithm (SSA) of Gillespie (1975) was used for the stochastic simulation. The steps below summarize the algorithm:

2) **Initialization (set initial numbers of species, set $t$=0, set stopping criteria):** Initialize the number of droplets in each species (the species are defined as droplets of different sizes). There is a unique index $\mu$ for each pair of droplets $i, j$ that may collide. For a system with $N$ species $\left(n_1, \quad n_2, \quad ... \quad ,n_N\right)$ $\mu \in \dfrac{N(N+1)}{2}$. The set $\{\mu\}$ defines the total collision space, and is equal to the total number of possible interactions.

3) **Monte Carlo step: D**etermine the next collision to occur and the time to the next collision. The next collision $\mu$ is calculated according to the distribution $P(\mu) = \dfrac{a_\mu}{\alpha}$, from the inequality:

$$\sum_{v=1}^{\mu-1} a_v < r_2\alpha \le \sum_{v=1}^{\mu} a_v \tag{B1}$$

Where $r_2$ is a uniformly distributed random number in the interval (0,1), $a_\mu$ are calculated from the probabilities:

$$a(i, j) = V^{-1}K(i, j)n_i n_j dt \equiv \Pr\{ \text{ Probability that two unlike particles } i \text{ and } j$$

with populations (number of particles) $n_i$ and $n_j$ will collide within the imminent time interval}

$$a(i,i) = V^{-1}K(i,i)\frac{n_i(n_i-1)}{2}dt \equiv \Pr\{ \text{Probability that two particles of the same}$$

species $i$ with population (number of particles) $n_i$ collide within the imminent time interval}

and $\alpha = \sum_{v=1}^{\frac{N(N+1)}{2}} a_v$ . As the time to the next collision is exponentially distributed with mean $1/\alpha$ (Gillespie, 1975), and that $1-r_1=r^*_1$ is a uniformly distributed random number in the interval [0, 1], then the time $\tau$ to the next collision can be calculated from the expression:

$$\tau = \frac{1}{\alpha}\ln\left(\frac{1}{r_1^*}\right) \qquad (B2)$$

5) Increase the time by the randomly generated time in Step 2. Change the numbers of species to reflect the execution of a collision.

6) If stopping criteria are not met, go to step 2.

**Anonymous Referee #1:**
*17. L167: Can you give a physical explanation about why you choose "C=5,49\*10^(10) cm^3s^{-1}"?*

**Reply to referee:**
**(This was discussed in the answer to question 15):**

The value of the constant $C$ ($C=5.49\times10^{10}$) in the product kernel $K(i,j) = Cx_ix_j$ is a result of the polynomial approximation (Long, 1974):
$$K(x,y) = A + B(x+y) + Cxy \qquad (S9)$$
of the hydrodynamic collection kernel:
$$K(x,y) = \pi[R(x)+r(y)]^2 E(x,y)[V(x)-V(y)] \qquad (S10)$$
Long (1974) calculated the coefficients for the polynomials (9) approximating the hydrodynamic kernel (10) when the largest of the colliding drops is smaller than 50 μm. The

results obtained by Long (1974) for the product kernel are displayed in Table S1(see the answer to question 15).

**Changes made in the revised manuscript:**

**Line 223 of the revised version:** The value of the constant C (C=5.49×10$^{10}$) in the product kernel is the result of the polynomial approximation $K(x,y) = A + B(x+y) + Cxy$ (Long, 1974) of the hydrodynamic collection kernel (Eq. 11).

**Anonymous Referee #1:**
*18. L173: Could you please explain more about the "mixing fraction", like mixing fraction of which quantity and the corresponding physical picture or intuition?*
**Reply:**
Looking for more clarity, we will change the terminology. In the equation (6) of the paper:

$$f(x,\theta,\mu_1,\beta,\mu_2,\sigma) = \theta Gumbel(x,\mu_1,\beta) + (1-\theta)Gauss(x,\mu_2,\sigma)$$

the coefficients $\theta$ and (1- $\theta$) are the probabilities associated with each component and are called the **mixture weights.** The **individual distributions** $Gumbel(x,\mu_1,\beta)$ and $Gauss(x,\mu_2,\sigma)$ that are combined to form the mixture, are the mixture components. We **will eliminate the term "mixing fraction"** in the revised version looking for more clarity.

**Changes made in the revised manuscript:**

The term mixing fraction was eliminated in the revised version. The terminology was changed.

**Line 181 of the revised version:**

$$f(x,\theta,\mu_1,\beta,\mu_2,\sigma) = \theta Gumbel(x,\mu_1,\beta) + (1-\theta)Gauss(x,\mu_2,\sigma) \tag{6}$$

In Eq. 6, the coefficients $\theta$ and (1- $\theta$) are the probabilities associated with each component and are called the mixture weights. The individual distributions $Gumbel(x,\mu_1,\beta)$ and $Gauss(x,\mu_2,\sigma)$ that are combined to form the mixture, are the mixture components.

**Anonymous referee #1:**

*20. L260: Please provide reference for "The block maxima (BM) approach in extreme value theory (EVT) was applied" and compare with the large deviation theory/method described in Wilkinson 16 [3].*

**Reply to referee:**

The main (classical) reference for extreme value theory (EVT) and the block maxima approach in is the book of **Gumbel (1958)**.

The large deviation theory (LDP) is concerned with the behavior of the tails of the distribution for the sum of independent and identical distributed (i.i.d) random variables. The Central Limit theorem is limited to values of the random variable not too far from the mean value. In the revised version, the difference between the Wilkinson's (2016) approach and ours will be discussed in both the introduction and conclusions. The discussion below will add more clarity to the discussion of the differences between the two approaches.

**Large deviation theory (LDT):** Suppose $X_1, X_2, ..., X_n$ is a sequence of independent, identically distributed random variables, with mean $\mu$ and variance $\sigma^2$, and $\bar{X}_n = \frac{1}{n}\sum_{i=1}^{n} X_i$, then:

$$\lim_{n \to \infty} \Pr\left(\frac{\bar{X}_n - \mu}{\sigma\sqrt{n}} \leq z\right) = \Phi(z) \tag{S10}$$

Then, for large $n$, the random variable $\frac{\bar{X}_n - \mu}{n\sqrt{\sigma}}$ have a Gaussian distribution with mean $\mu=0$ ans standard deviation 1. This is the Central Limit theorem (CLT). Then, the CLT says that

$$\lim_{n \to \infty} \Pr\left(\bar{X}_n \leq \mu + z\sigma\sqrt{n}\right) = \Phi(z) \tag{S11}$$

And

$$\lim_{n \to \infty} \Pr\left(\bar{X}_n \geq \mu + z\sigma\sqrt{n}\right) = 1 - \Phi(z) \tag{S12}$$

However, we must take into account that the range of validity of the CLT is:

$$|x - \mu| = O\left(\frac{1}{\sqrt{n}}\right) \tag{S12}$$

And as a consequence, is not very accurate in the tails of the distribution. For example, if we approximate the tail probability $\Pr\left(\bar{X}_n \geq \mu + \varepsilon\right)$ for a fixed value of $n$ by using (S12), the result will be not very accurate if $\bar{X}_n$ is far from the mean, unless $n$ is sufficiently large. Then, we need an expression more accurate than the Gaussian distribution for finite but large values of $n$, and that will recover the Gaussian distribution when $n \to \infty$. The large deviation theory

provides a solution for this problem. **Summarizing, the large deviation theory (LDP) is concerned with the behavior of the tails of the distribution. The Central Limit theorem is limited to values of the random variable not too far from the mean value. Then, according to the large deviation theory for sums of independent random variables:**

$$f(x) \rightarrow C(n)\,\mathrm{e}^{-nI(x)} \tag{S13}$$

Where $C(n)$ is a normalizing constant, and $I(x)$ is the large deviation function. The distribution (S13). This expression is accurate for all values of *x,* while *the* Gaussian is accurate only for values that fulfilled the condition $|x - \mu| = O\!\left(\dfrac{1}{\sqrt{n}}\right)$.

**Wilkinson (2016) application of large deviation theory (LDT)**: Due to the fact that the precipitation occurs on a time scale that is smaller than the typical scale for one collision, the problem can be solve by applying large deviation theory (Wilkinson, 2016). Then, it is necessary to determine the probability density for the time $T_{N_c} = \sum\limits_{i=1}^{N_c} t_i$ being a small fraction of its average value (far from the mean value). In the former expression the $t_i$ are the times between droplet collisions and $N_c$ the number of collisions, which have an exponential distribution (Wilkinson, 2016). Then, we need to find the probability at the tails of the distribution. According to LDT, the probability can be written in the form:

$$f(\tau) = \frac{1}{\langle T \rangle}\,\mathrm{e}^{-J(\tau)} \tag{S14}$$

where $J(\tau)$ is the rate function (see Eq. S13) that was explicitly calculated in Wilkinson's (2016) paper.

**This paper's approach:** The approach we follow in this report is different, since we are interested on finding the size distribution of runaway droplets that trigger precipitation formation.

**References:**
Gumbel, E.J.: Statistics of Extremes. Columbia University Press, 1958 - 375 pp.
Wilkinson M. 2016. Large deviation analysis of rapid onset of rain showers. Phys. Rev.
      Lett. 116: 018 501, doi:10.1103/PhysRevLett.116.018501.

**Changes made in the revised manuscript:**

The large deviation theory was discussed in detailed in the reply to the reviewer, but only briefly mentioned as an alternative method to explore the sol-gel transition.

**Line 83 of the revised version:** More recently, Wilkinson (2016) advanced further the model

by using large deviation theory (Touchette, 2009). He derived the probability for the time T

to undergo N collisions being a very small fraction of its mean value, and showed that the time scale for the initiation of precipitation is smaller than the mean time for a single collision.

**Reviewer:**
*21. L271-272: Please rephrase the sentence "The sample size...of data" to improve the readability. The "which clause" is not encouraged in scientific writing.*
**Reply:**
In the revised version the sentence will be rewritten as suggested.

**The corresponding modifications were made.**

**Reviewer:**
*22. L318: Did you mean "entire dataset"?*
**Reply:**
Yes, the "entire dataset". The corresponding change will be made.

**The corresponding modifications were made.**

**Section 2: Response to Referee # 2 comments and modifications made to the manuscript (marked in red in the revised manuscript):**

1) **Anonymous Referee #2:**

*My primary contention is that the authors don't show why the said hypothesis (authors call it sol-gel hypothesis) will lead to a particular frequency distribution of the size of the biggest droplets or the shown size distribution is unique to only this hypothesis. Second, the connection between observed frequency distribution in fog data and simulations is tenious.*

**Reply to reviewer:**

In Botet and Płoszajczak (2005), the Kolmogorov-Smirnov distribution was found as the exact largest cluster distribution for critical mean-field percolation. In the same paper, by means of detailed numerical simulations of the Smoluchowski equation with a product kernel, the authors conjectured that the Kolmogorov-Smirnov distribution is the largest particle distribution at sol-gel transition time. This results are for infinite systems.

To the best of my knowledge, there is no similar theoretical result for the largest cluster distribution at critical point in finite systems. For finite systems, in a previous paper (Botet, 2011), it was demonstrated (by means of detailed numerical simulations for multiplicative

coagulation), that the distribution (1) is a good approximation to the empirical largest particle distribution for all values of $\theta$.

$$f(x,\theta,\mu_1,\beta,\mu_2,\sigma) = \theta Gumbel(x,\mu_1,\beta) + (1-\theta)Gauss(x,\mu_2,\sigma) \qquad (1)$$

Our approach is based on numerical simulations, and analysis of observations in order to check whether or not the largest droplet radius will follow a mixture of Gumbel and gaussian distributions. One aspect that strengthens our hypothesis is that we used formal statistical tests in order to check whether or not the observed distribution follow or not the mixture of distributions, with good results. These points are analyzed in more detailed below.

**Results for finite systems:**

It was hypothesized that the largest droplet mass (for finite systems) is distributed following an admixture of distributions (Gumbel and Gaussian), because in an early stage of cloud formation (disordered or statistical phase), fluctuations and correlations are negligible, there only a few collision events, and droplets are randomly distributed. According to extreme value theory, in a system without correlations, the random variable $s_{max} = \max\limits_{i=1,...,m} s_i$ , where $s_i$ denotes the size of the droplet number $i$, and there are $m$ droplets in the system), is distributed following an asymptotic $\Phi_{max}(s_{max})$ , which must be either Gumbel, Frechet or Weibull distribution (Gumbel, 1958). Examples of distributions belonging to the domain of attraction of the Gumbel distribution are the exponential and the Gaussian distribution, a fact that explain the ubiquity of the Gumbel distribution in extreme value theory applications.

For an uncorrelated system, no other distributions can appear for the extremal variable. At later times, away from the pseudocritical region (which is the finite system equivalent of a sol-gel transition time) the Central Limit theorem applies, and the corresponding probability distribution must be Gaussian (provided correlations be negligible). The equation (1) will work for the uncorrelated case (early stage of collision-coalescence process), but with small (close to 0) or large (close to 1) values of $\theta$. In the presence of correlations, in the pseudocritical region (which is the equivalent of a phase transition for a finite system) the values of $\theta$ will be in the vicinity of 0.5, and $f(x,\theta,\mu_1,\beta,\mu_2,\sigma)$ takes a non-trivial form, which is well fitted by Eq. (1).

In our paper, the Kolmogorov-Smirnov goodness of fit test were performed in order to check whether or not the distribution (1) fits our data. The admixture of distributions was always a better model than the Gumbel and Gaussian distributions for all sample sizes, with large p-values. So, in the majority of the cases, the null hypothesis (the sample comes from a mixture of distributions) could not be rejected.

Gruyer et al. (2013), in a very different context, found that the distribution of the largest fragments in nuclear multifragmentation reactions (simultaneous break-up of an excited nuclear system in several fragments) have o nontrivial form which is well fitted by the function (1). The context is different, but the physical interpretation is similar. At higher and lower energies, fluctuations are negligible, and the largest fragment is distributed Gumbel or Gaussian. At intermediate energies (with larger correlation), there is an admixture an admixture of the two asymptotic distributions.

**Results for infinite systems:**

In theory of critical phenomena, little is known about the distribution of the largest fragment at the critical point. However, there is a remarkable result in mean-field percolation theory (Botet and Płoszajczak, 2005), showing (from theoretical backgrounds), that the largest cluster distribution at the critical point is the Kolmogorov-Smirnov (K-S) distribution. The K-S distributionis the distribution of the maximum value of the deviation between the experimental realization of a random process and its theoretical cumulative distribution and it has the cumulative distribution:

$$K_1(z) = \sum_{k=-\infty}^{\infty} (-1)^k e^{-k^2\pi^2 z/6} \qquad (2)$$

Or the equivalent expression:

$$K_1(z) = \sqrt{\frac{6}{\pi z}} \sum_{k=-\infty}^{\infty} e^{-3(2k+1)^2/(2z)} \qquad (3)$$

In Botet and Płoszajczak, 2005 it was also conjectured, by means of numerical simulations, that the largest particle distribution at sol-gel transition time for the Smoluchowski model (Eq. 4) with a product kernel $K(i,j)=Cx_i x_j$ is also the Kolmogorov-Smirnov distribution (1,2).

$$\frac{\partial N(i,t)}{\partial t} = \frac{1}{2} \sum_{j=1}^{i-1} K(i\text{-}j,j)N(i\text{-}j)N(j) - N(i) \sum_{j=1}^{\infty} K(i,j)N(j) \qquad (4)$$

We must remember that, for equation (4), there is a sol-gel transition at $T_{gel} = \left[ CM_2(t_0) \right]^{-1}$,

where $\quad M_2(t) = \sum_{i=1}^{N_d} x_i^2 N(i,t) \quad$ is the second moment of droplet mass spectrum.

**2) Referee Comment:**

*Third, sufficient details are not provided to ascertain whether the frequency distribution as the sum of two distributions with a particular value of weighting factor is better than say if only one of the distribution (that is weighting factor either zero or one) was fitted.*

**Reply to referee:**

The results with the Kolmogorov-Smirnov goodness of fit test are shown in Table 1 of the paper. As can be observed in Table 1, for 100 randomly selected samples with different sample sizes, the admixture is always a better model than the Gumbel and Gaussian distributions for all sample sizes.

The fact that the Gumbel distribution fits so poorly the data is because most of the samples are in the pseudo-critical region. Effectively, Figure 8 of the paper shows that for 90% of the samples the ratio $\eta = \dfrac{w_{Gaussian} - w_{Gumbel}}{w_{Gaussian} + w_{Gumbel}}$ lies in the interval [-0.9, 0.9], clearly indicating that 90% of the samples are on the pseudocritical domain.

Table 1. For each sample size, number of samples with the null hypothesis $H_0$ rejected at α=0.05 for all the distributions.

| Case | Total number of random samples | Sample size | Fitted Distributions | At α=0.05 Reject $H_0$ (Number of Samples) |
|---|---|---|---|---|
| 1 | 100 | 100 | Mixture | 13 |
| | | | Gumbel | 92 |
| | | | Gaussian | 35 |
| 2 | 100 | 200 | Mixture | 27 |
| | | | Gumbel | 96 |
| | | | Gaussian | 58 |
| 3 | 100 | 300 | Mixture | 35 |
| | | | Gumbel | 98 |
| | | | Gaussian | 70 |
| 4 | 100 | 400 | Mixture | 40 |
| | | | Gumbel | 100 |
| | | | Gaussian | 77 |
| 5 | 100 | 500 | Mixture | 50 |
| | | | Gumbel | 100 |
| | | | Gaussian | 83 |

In the revised version we will make more emphasis on this problem.

**References:**

Botet, R.: Where are correlations hidden in the distribution of the largest fragment?. *PoS*, 007, 2011.

Botet, R. and Płoszajczak, M.: Exact order-parameter distribution for critical mean-field percolation and critical aggregation. *Physical Review Letters*, 95(18), 185702, 2005.

Gruyer, D., Frankland, J. D., Botet, R., Płoszajczak, M., Bonnet, E., Chbihi, A., ... and Guinet, D.: Nuclear multifragmentation time scale and fluctuations of the largest fragment size. *Physical review letters*, 110(17), 172701, 2013.

**Changes made in the revised manuscript:**

The correctness of the choice of the admixture was demonstrated through a formal statistical analysis, and the fact that the mixture of distributions is a better model that the Gumbel and Gaussian was already stated in the original version of the paper.

**Line 395 of the revised paper:** The results shown in Table 1, confirm that for all sample sizes, the mixture of distributions provides a better fit than the Gumbel and Gaussian distributions, confirming the correctness of the choice of the mixture of distributions (Eq. 6) for modelling the largest droplet radius. As an example, Figs. 7a-d present, for a sample size of $n$=500, the largest droplet mass empirical distributions obtained for four different samples that are distributed following the mixture, and the corresponding fit of Eq. 6.